



# Satellite-based estimation of roughness lengths and displacement heights for wind resource modelling

Rogier Floors[1], Merete Badger[1], Ib Troen[1], Kenneth Grogan[2], and Finn-Hendrik Permien[3]

[1]Technical University of Denmark, DTU Wind Energy, Risø Campus, Frederiksborgvej 399, 4000 Roskilde, Denmark
[2]DHI GRAS A/S, Agern Alle 5, 2970 Hørsholm, Denmark
[3]Siemens Gamesa Renewable Energy A/S, Borupvej 16, 7330 Brande, Denmark

**Correspondence:** Rogier Floors (rofl@dtu.dk)

**Abstract.** Wind turbines in northern Europe are frequently placed in forests, which sets new wind resource modelling requirements. Accurate mapping of the land surface can be challenging at forested sites due to sudden transitions between patches with very different aerodynamic properties, e.g. tall trees, clearings, and lakes. Tree growth and deforestation can lead to temporal changes of the forest. Global or pan-European land cover data sets fail to resolve these forest properties, aerial lidar
campaigns are costly and infrequent, and hand-digitization is labour-intensive and subjective. Here, we investigate the potential of using satellite observations to characterise the land surface in connection with wind energy flow modelling using the Wind Atlas Analysis and Application Program (WAsP). Collocated maps of the land cover, tree height, and Leaf Area Index (LAI) have been generated based on observations from the Sentinel-1 and -2 missions combined with the Ice, Cloud, and Land Elevation Satellite-2 (ICESat-2). Three different forest canopy models are applied to convert these maps to roughness
lengths and displacement heights. We introduce a modified model, which can process detailed land cover maps containing both roughness lengths and displacement heights. Extensive validation is carried out through cross-prediction analyses at ten well-instrumented sites in various landscapes. We demonstrate that using the novel satellite-based input maps leads to lower cross-prediction errors of the wind power density than land cover databases at a coarser spatial resolution. Differences in the cross-predictions resulting from the three different canopy models are minor. The satellite-based maps show cross-prediction
errors close to those obtained from aerial lidar scans and hand-digitised maps. This demonstrates the value of using detailed satellite-based land cover maps for micro-scale flow modelling.

## 1  Introduction

Wind turbines on land represent a cost-competitive source of renewable energy (Global Wind Energy Council, 2019). More than 95% of the global installed wind energy capacity of $\approx 651$ GW in 2019 is installed on land (https://www.irena.org/wind).
In the northern Europe's temperate climates, a vast amount of the land surface is covered by forest. The exploitation of wind power within such forests has become more widespread as the hub height of modern wind turbines has exceeded the forest height (Enevoldsen, 2016).

Wind resource assessment is typically performed with linearized modelling in a wind farm siting tool (e.g. WAsP or Wind-PRO), which contains several sub-models to predict the flow based on an input wind histogram (Troen and Petersen, 1989).



Based on Monin-Obukhov similarity theory (Eq. 1), the flow model can be used to predict the wind speed, $U$, for any height, $z$ above the ground:

$$U = \frac{u_*}{\kappa} \left[ \ln \frac{z-d}{z_0} - \Psi_m \right], \tag{1}$$

where $u_*$ is the friction velocity, $\kappa$ is the Von Kármán constant ($= 0.4$), $z_0$ is the aerodynamic roughness length, and $\Psi_m$ represents the stability correction to the profile and depends on $z/L$, where $L$ is the Monin-Obukhov length (Businger et al., 1971). At forested sites, the zero-plane displacement height, $d$ is traditionally used in addition to $z_0$ to account for the canopy forcing the mean flow to be displaced upwards over forests (Thom, 1971). A dense forest appears more smooth (i.e. a lower $z_0$) and has a larger $d$ than a sparse forest with clearings and a more in-homogeneous appearance of the same height (Shaw and Pereira, 1982).

Values of $z_0$ and $d$ can be assessed through manual inspection of the forest at a given site combined with digital maps. In practice, a background value for $z_0$ and $d$ is set, and adjustments are made for specific areas where the roughness and displacement height differ from the background values. Manual assessments are subjective and time-consuming, and they can lead to a high level of uncertainty of the estimated wind resource (Kelly and Jørgensen, 2017).

Fully automated assessments of $z_0$ and $d$ can be achieved based on global or regional land cover data sets derived from satellite observations. Each land cover class is assigned a value of $z_0$ and $d$ via a land cover table (Jancewicz and Szymanowski, 2017; Floors et al., 2018). State-of-the-art flow modelling tools offer embedded access to such land cover maps and to the associated roughness translation tables, which the user may modify. Due to the coarse spatial resolution of global land cover data sets and the typical minimum-mapping-unit (MMU) of several hectares, the finer-scale variability within a forest, such as smaller clearings, is not resolved. Further, the available land cover to roughness translation tables may not be fully representative for the site in question due to the data sets' global nature. Commonly used tables show very low roughness lengths for forest land cover classes (Floors et al., 2018; Enevoldsen, 2016).

Floors et al. (2018) have demonstrated that tree heights and forest densities retrieved from aerial lidar scans can be used to parameterize $z_0$ and $d$ over the forest. This approach is more physical than the ad-hoc assignment using land cover data sets. It sets new requirements to the flow modelling tools used for wind energy siting because i) a different type of model is needed to estimate $z_0$ and $d$ from the tree height and density, and ii) efficient data processing routines are necessary to handle the much finer resolution of data layers from aerial lidar scans. Several countries in northern Europe have released national aerial lidar scans, and dedicated campaigns may be performed in connection with wind farm planning. A limitation of the aerial lidar scans is each campaign's high cost and the associated data processing. Therefore, the temporal frequency of such observations is low.

A wealth of new satellite observations with unprecedented spectral properties and spatial and temporal resolutions have become available, e.g. through Copernicus (https://www.copernicus.eu). Forest monitoring is a key objective of several new missions because information on deforestation and forest degradation is important in connection with climate change mitigation. However, key metrics for wind-resource assessment such as forestry canopy height are still missing or only available at a low spatial resolution but can be derived through post-processing of the data from available sensors. Here, we hypothesize that





these post-processed values of forest canopy height and density retrieved from satellites in a high spatial resolution, can also be used to estimate wind resources in the same accuracy as aerial lidar scans but at a lower cost.

This paper aims to use tree heights and densities retrieved from satellite observations in connection with flow modelling for wind energy. We will map $z_0$ and $d$ from satellite-based forest parameters and use these maps in combination with wind observations from meteorological masts to predict the wind resource for ten sites worldwide. Wind power densities calculated using the novel satellite-based data layers will be compared to predictions based on global land cover maps, aerial lidar scans, and manual digitization for the parameterization of $z_0$ and $d$.

## 2   Background

### 2.1   Forest parameters from satellites

The Ice, Cloud, and Land Elevation Satellite-2 (ICESat-2) carries the Advanced Topographic Laser Altimeter System (ATLAS), used to observe the land surface height with great precision. Amongst many other parameters, the mission delivers global forest canopy heights (Neuenschwander and Pitts, 2019), which are well correlated with canopy heights from aerial lidar scans (Li

et al., 2020) and field measurements (Huang et al., 2019). A mission with similar capabilities to ICESat-2 is the Global Ecosystem Dynamics Investigation (GEDI). However GEDI only collects data 52 degrees north and south of the equator, and therefore does not offer global coverage. Although representing a major advancement in estimating 3D forest structure, these current spaceborne laser observations are restricted to very narrow footprints, which are insufficient to map wall-to-wall canopy heights over larger areas. Recent studies have shown that the laser-derived canopy heights can be extrapolated

through different combinations with other satellite observations and machine learning techniques (Csillik et al., 2020; Fagua et al., 2019). Typically, the canopy heights from laser measurements are extrapolated using textural information from active microwave sensors (e.g. Sentinel-1 or ALOS PALSAR) and multispectral information from passive sensors (e.g. Landsat or Sentinel-2). Li et al. (2020) has pointed out that backscattering coefficients from Sentinel-1 and the variables related to red-edge bands from Sentinel-2 contribute positively to the prediction of forest canopy heights.

The Leaf Area Index (LAI) is a measure of the one-sided green leaf area per unit ground area in broadleaf canopies and as one-half the total needle surface area per unit ground area in coniferous canopies (Chen and Black, 1992). In connection with flow modelling for wind energy, LAI can be used as a proxy for the forest density (Raupach, 1994). Several space-borne sensors, operating in the visible and near-infrared range, monitor vegetation properties (including LAI) daily. For example, the Terra and Aqua satellites each carry a Moderate Resolution Imaging Spectroradiomenter (MODIS). Four-day composites of

LAI are generated routinely from the two instruments in combination. This product has a relatively coarse resolution and a pixel size of 500 m. Guzinski and Nieto (2019) has developed a method for downscaling of LAI estimates.



## 2.2 Forest roughness models

Different models can be used to estimate $z0$ and $d$ from forest canopy heights and densities. Here, we consider the Objective Roughness Approach (ORA), the Raupach model, and the scalar distribution (SCADIS) model.

### 2.2.1 The objective roughness approach (ORA)

The relation between $h$ and $z_0$ and $d$ has been recognized and discussed by many authors (Thom, 1971; De Bruin and Moore, 1985). Because $z_0$ is usually proportional to $h$, an easy way to obtain $z_0$ and $d$ is by relating them linearly to the canopy height,

$$z_0 = c_1 h, \tag{2}$$

and

$$d = c_2 h. \tag{3}$$

The constant values $c_1 = 0.1$ and $c_2 = 2/3$, were shown in Floors et al. (2018) to yield good results when used for wind energy flow modelling at a forested site.

### 2.2.2 The Raupach model

Raupach (1992) developed a model to predict the bulk drag coefficient over a rough surface with a canopy height. The main model parameter resulting from his analysis is the frontal area index $\lambda$, which for isotropically oriented elements is given by,

$$\lambda = 0.5\text{LAI}. \tag{4}$$

To a good approximation, the LAI can be substituted by the canopy area index, which includes the area covered by the canopy objects that are not leaves (Raupach, 1994). For simplicity, we here refer to the more commonly used LAI throughout the paper. Raupach (1994) discussed some simplifications to the original model and suggested

$$\frac{d}{h} = 1 - b, \tag{5}$$

where

$$b = 1 - \frac{\exp(-a)}{a} \tag{6}$$

and

$$a = \sqrt{2c_{d1}\lambda}. \tag{7}$$

$c_{d1}$ was experimentally found to be equal to 7.5. Finally, the roughness length is estimated as,

$$\frac{z_0}{h} = b\exp\left(\frac{-\kappa}{\min(\sqrt{C_S + C_R\lambda}, c_{\max})} - \Psi_h\right). \tag{8}$$

Raupach (1994) suggested the constants to be $C_S = 0.003$, $C_R = 0.3$, $c_{\max} = 0.3$ and $\Psi_h = 0.193$.





### 2.2.3 The SCADIS model

A criticism of the Raupach (1994) model is that some of the constants are highly dependent on the structure of the canopy and may not be universally applicable. To address this, one can use a one-dimensional version of a $k$-$\omega$ model, which depends on the leaf-area density distribution within the canopy. Such an approach was suggested by Sogachev et al. (2002); Sogachev and Panferov (2006). An additional advantage of this model is that it can estimate $z_0$ and $d$ in non-neutral atmospheric conditions.

The leaf or canopy area density (LAD) profile is frequently described (e.g. Meyers and Tha Paw U, 1986) by a beta proba-
bility density function,

$$\text{LAD} = \text{LAI}\left[\left(z/h\right)^{\alpha-1}\left(1-z/h\right)^{\beta-1}\right], \tag{9}$$

where $\alpha = 9$ and $\beta = 3$. The chosen constants are representative of a temperate pine forest. Sogachev et al. (2017) validated the SCADIS model for flow over a 3D area with forested terrain and found that simulations that explicitly resolve the drag of the canopy and those that use an effective $z_0$ compare well.

### 2.3 Model response to LAI

The behaviour of the three forest roughness models for a canopy height $h$ of 10 m as a function of LAI is shown in Fig. 1. The ORA model does not depend on leaf-area index and is therefore constant, i.e. $d/h = 2/3$ and $z_0/h = 0.1$. The Raupach and SCADIS models both show an increasing $d$ as the LAI increases. $z_0$ has the opposite behaviour and decreases to lower values after an initial maximum of around LAI$\approx$1.

The main differences between the SCADIS and Raupach models occur for a relatively low LAI $\sim 1$; using the SCADIS model $z_0/h \approx 0.2$ while using the Raupach model $z_0/h \approx 0.1$. The same holds for $d$, where using the Raupach model yields a $d$ that is nearly twice that of the SCADIS model. For the more commonly occurring LAI$> 3$, the differences are minor. When a different canopy profile is specified, i.e. using $\alpha = 5$ and $\beta = 3$, there are larger differences between the two models.

### 2.4 Flow modelling and cross-prediction

Maps of $z_0$ and $d$ can be used in combination with terrain elevation maps as input to wind energy flow modelling. The WAsP methodology (Troen and Petersen, 1989) can be applied to analyze the observed wind climate (a sector-wise histogram of wind speeds) at a particular mast and height and predict the wind climate at a nearby location, assuming the same large-scale atmospheric forcing. Speed-ups caused by the land surface roughness and orography are first 'subtracted' from the observed wind climate. Eq. 1 and the geostrophic drag law (Blackadar and Tennekes, 1968) are then used to estimate the geostrophic
wind speed distribution, which is assumed to be valid for a larger area surrounding the mast. It can be transformed to Weibull distributions over idealized flat terrain at a number of specified heights and $z_0$s (see Sect. 8.7 in (Troen and Petersen, 1989)). The result of this process is a so-called Generalized Wind Climate (GWC) object, which contains the Weibull parameters $A$ and $k$ and the probability density per wind direction sector. The GWC can be used to predict the wind resource near the measurement site by 'adding' local roughness and orographic speed-up factors. The process of predicting the wind climate

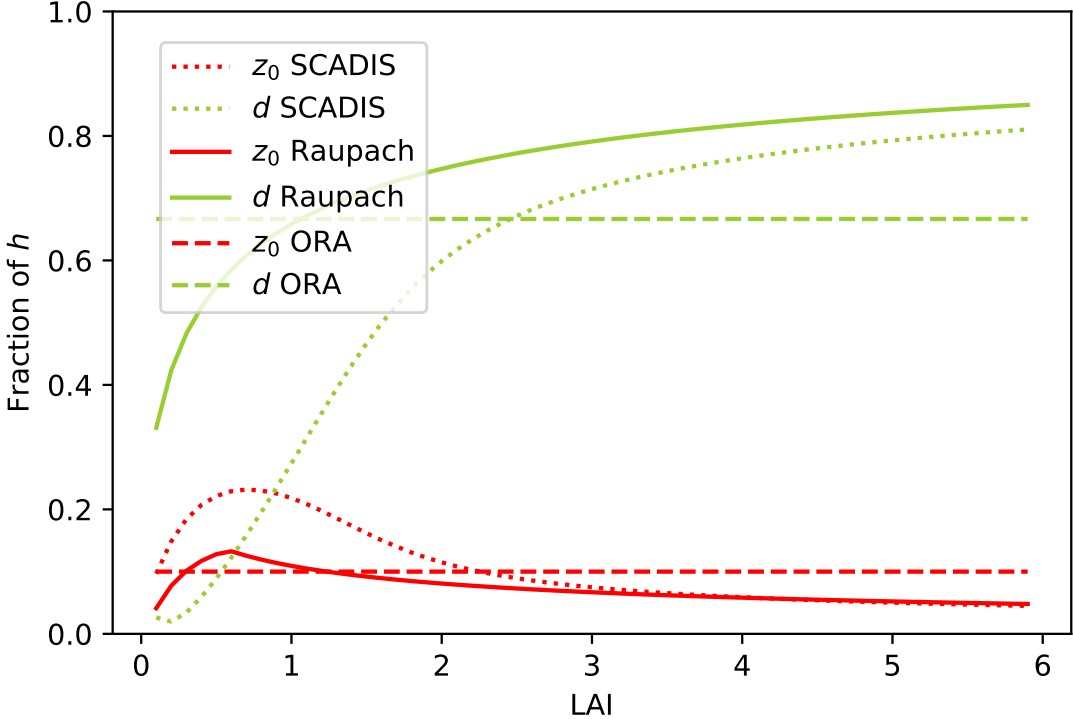

**Figure 1.** Response of the different roughness models to leaf-area index (LAI). The SCADIS model is set up with $\alpha = 9$ and $\beta = 3$, which are typical values for forests with most of the canopy density in the upper part of the canopy layer (Sogachev et al., 2017).

from one position to another is called a cross-prediction. The accuracy of such predictions depends strongly on the quality of the roughness and elevation maps used as input (Floors et al., 2018).

## 3   Sites and data

In this section, we present ten sites that are considered in this study and the different types of input data, which are needed for cross-prediction analyses at the sites.

### 3.1   Measurement sites

Ten global measurement sites are selected for cross-prediction analyses (Table 1). The sites represent vastly different land cover types and complexities in terms of elevation changes. The sites Cuauhtémoc and Humansdorp are surrounded by very open landscapes where the WAsP model is expected to perform well. The sites Risø (Giebel and Gryning, 2004) and Østerild (Peña, 2019) are characterized by a mixture of forest and open areas, which is more challenging in terms of flow modelling. The sites

Ryningsnäs, Finland, Sweden and Mérida, are located in forests, where the displacement height becomes important, and the uncertainty in wind resource assessment is known to be high (Kelly and Jørgensen, 2017). The sites in Sweden and Finland are





from confidential projects, and the exact locations of these can therefore not be disclosed. Perdigao and Alaiz are complex sites inmountainous terrain. The ruggedness index (RIX) is often used to characterize the flow modelling complexity. It indicates the percentage of slopes higher than 30% in a 3.5 km circle around the site. For larger values of this index, WAsP has been
found to become inaccurate (Bowen and Mortensen, 1996). This is probably due to increased form drag on steep slopes, not accounted for in the surface roughness. In addition, nearby steep slopes are not well handled in the BZ terrain model in WAsP (Troen and Hansen, 2015).

### 3.2 Standard land cover data sets

Four course-resolution land cover databases that are regularly applied for wind energy modelling are used in connection with
this analysis (see Table 3).

The Global Land Cover Characterization (GLCC) dataset provides a coarse 1000 m resolution land cover data set. It is derived from the Advanced Very High Resolution Radiometer (AVHRR) data collected between April 1992 and March 1993. The AVHHR is a space-borne sensor mounted on a polar-orbiting satellite from the National Oceanic and Atmospheric Administration (NOAA). The MODIS (MCD12Q1 V6) product provides yearly global land cover between 2001 and 2018 and is derived
from six different classification schemes. It is made using supervised classifications of MODIS Terra and Aqua reflectance data. Prior knowledge and ancillary information is used to refine the classifications further. This product has a resolution of $\approx 500$ m. The C3S Global Land Cover product provides global coverage at a resolution of $0.002778° (\approx 300$ m$)$. It is consistent with the global annual land cover series from 1992 to 2015 produced by the European Space Agency (ESA) Climate Change Initiative (CCI). Version 2.1.1 created from data from 2018 is used here, which has 38 land cover classes. The most recent
version (2018) of the Corine Land Cover (CL2018) inventory is used to analyse sites in Europe. Its minimum mapping unit is 25 ha, corresponding to a grid resolution of 100 m, and the product has 44 land cover classes.

The coarse-resolution land cover data sets come with standard roughness conversion tables (Table A1–A5, Thøgersen (2021)). The $z_0$ for forest classes tends to be too low for temperate forests (Floors et al., 2018). Recent studies of Dörenkämper et al. (2020) and Badger et al. (2015) have highlighted this aspect and have suggested higher $z_0$ for forested areas. To reflect
the uncertainty of $z_0$ assignments, we include these in Table A3–A5. In connection with the C3S Global Land Cover product, the $z_0$ classification is based on the data set of 2009 and some of the 23 land cover classes have been split into sub-classes since then. Therefore, $z_0$ of each subclass is assumed to be identical to $z_0$ of the class it inherits from (see Table A3).

### 3.3 Novel Sentinel data sets

High-resolution satellite-based data packages have been developed for an area of $40 \times 40$ km around each of the ten sites
described above. Each data package has a regular grid spacing of 20 m by 20 m and includes 1) land cover classification, 2) LAI, and 3) forest canopy height. The primary data source used for the production of these layers is Copernicus satellite imagery from the Sentinel-1 and Sentinel-2 missions. Sentinel-1 is a $c$-band (5.6 cm) Synthetic Aperture Radar sensor, while Sentinel-2 is an optical sensor providing data in the visible, near-infrared, and shortwave infrared parts of the spectrum. The land cover classification is based on five land cover classes most relevant for wind modelling (see Table A5). For each site,





**Table 1.** Sites and masts used for cross-prediction analyses. The characteristic land cover and terrain complexity (RIX) are indicated. The EPSG code identifies the coordinate reference system, including the map projection, datum and zone number (see text) and the $x$, $y$ and $z$ columns identify the positions in projected map coordinates.

| Site | Mast | Land cover type | RIX (%) | EPSG code | $x$ (m) | $y$ (m) | $z$ (m) |
|---|---|---|---|---|---|---|---|
| Alaiz (ES) | 1 | Complex | 9.5 | 32630 | 617609 | 4731301 | 20.0, 40.0, 60.0, 80.0 |
| Alaiz (ES) | 2 | Complex | 8.0 | 32630 | 617650 | 4732567 | 20.0, 40.0, 60.0, 80.0 |
| Cuauhtémoc (MX) | | Open | 0.0 | 32613 | 309873 | 3211836 | 20.0, 40.0, 60.0, 80.0 |
| Finland (FI) | 1 | Forest | 0.0 | 32635 | | | 60.0, 90.0, 116.5 |
| Finland (FI) | 2 | Forest | 0.0 | 32635 | | | 61.0, 91.0, 116.5 |
| Finland (FI) | 3 | Forest | 0.0 | 32635 | | | 61.0, 91.0, 116.5 |
| Humansdorp (SA) | | Open | 0.0 | 32735 | 270726 | 6222861 | 20.2, 37.3, 61.1, 62.0 |
| Mérida (MX) | | Forest | 0.0 | 32616 | 210700 | 2339899 | 20.0, 40.0, 60.0, 80.0 |
| Perdigão (PT) | 1 | Complex | 17.2 | 32629 | 607386 | 4396302 | 30.0, 60.0 |
| Perdigão (PT) | 2 | Complex | 15.5 | 32629 | 607695 | 4395878 | 28.0, 57.0, 100.0 |
| Perdigão (PT) | 3 | Complex | 15.6 | 32629 | 607874 | 4395628 | 30.0, 60.0 |
| Perdigão (PT) | 4 | Complex | 18.1 | 32629 | 607877 | 4396972 | 60.0 |
| Perdigão (PT) | 5 | Complex | 16.5 | 32629 | 607934 | 4396109 | 30.0, 60.0 |
| Perdigão (PT) | 6 | Complex | 16.6 | 32629 | 608239 | 4397251 | 30.0, 60.0 |
| Perdigão (PT) | 7 | Complex | 15.3 | 32629 | 608449 | 4396471 | 30.0, 60.0, 100.0 |
| Perdigão (PT) | 8 | Complex | 15.7 | 32629 | 608628 | 4396602 | 30.0, 60.0 |
| Perdigão (PT) | 9 | Complex | 15.5 | 32629 | 608827 | 4396741 | 30.0, 60.0, 100.0 |
| Risø (DK) | | Mixed | 0.2 | 32632 | 694096 | 6176367 | 44.2, 76.6, 94.0, 118.0, 125.2 |
| Ryningsnäs (SE) | | Forest | 0.0 | 32633 | 559487 | 6348565 | 40.0, 59.0, 80.0, 98.0, 120.0, 138.0 |
| Sweden (SE) | 1 | Forest | 0.1 | 32633 | | | 57.8, 80.7, 96.4, 100.7 |
| Sweden (SE) | 2 | Forest | 0.2 | 32633 | | | 31.5, 44.5, 57.0, 59.0 |
| Sweden (SE) | 3 | Forest | 0.3 | 32633 | | | 57.8, 80.9, 96.4, 100.8 |
| Sweden (SE) | 4 | Forest | 0.0 | 32633 | | | 57.7, 80.8, 96.4, 100.8 |
| Sweden (SE) | 5 | Forest | 0.1 | 32633 | | | 32.1, 44.0, 57.3, 59.0 |
| Sweden (SE) | 6 | Forest | 0.3 | 32633 | | | 57.6, 96.4, 100.8 |
| Sweden (SE) | 7 | Forest | 0.2 | 32633 | | | 57.8, 80.9, 96.4, 100.8 |
| Østerild (DK) | 1 | Mixed | 0.1 | 32632 | 492766 | 6327084 | 40.0, 70.0, 106.0, 140.0, 178.0 |
| Østerild (DK) | 2 | Mixed | 0.0 | 32632 | 492767 | 6322834 | 40.0, 70.0, 106.0, 140.0, 178.0 |



**Table 2.** Overview of the input data that are used. The meaning of the different abbreviations are discussed in the text.

| Site | Elevation data | GLCC 1000 | MODIS 500 | GLOB 300 | CORINE 100 | Sentinel ORA 20 | Sentinel Raupach 20 | Sentinel SCADIS 20 | Hand digitized | Lidar scans ORA 20 |
|---|---|---|---|---|---|---|---|---|---|---|
| Alaiz (ES) | Lidar scans | ✓ | ✓ | ✓ | ✓ | ✓ | ✓ | ✓ | ✓ | |
| Cuauhtémoc (MX) | SRTM version 3 | ✓ | ✓ | ✓ | | ✓ | ✓ | ✓ | ✓ | |
| Finland (FI) | Lidar scans | ✓ | ✓ | ✓ | ✓ | ✓ | ✓ | ✓ | | ✓ |
| Humansdorp (SA) | SRTM version 3 | ✓ | ✓ | ✓ | | ✓ | ✓ | ✓ | ✓ | |
| Mérida (MX) | SRTM version 3 | ✓ | ✓ | ✓ | | ✓ | ✓ | ✓ | ✓ | |
| Perdigão (PT) | Lidar scans | ✓ | ✓ | ✓ | ✓ | ✓ | ✓ | ✓ | | |
| Risø (DK) | Lidar scans | ✓ | ✓ | ✓ | ✓ | ✓ | ✓ | ✓ | ✓ | |
| Ryningsnäs (SE) | Lidar scans | ✓ | ✓ | ✓ | ✓ | ✓ | ✓ | ✓ | | ✓ |
| Sweden (SE) | Lidar scans | ✓ | ✓ | ✓ | ✓ | ✓ | ✓ | ✓ | | ✓ |
| Østerild (DK) | Lidar scans | ✓ | ✓ | ✓ | ✓ | ✓ | ✓ | ✓ | | ✓ |

**Table 3.** Summary of the different land cover data sources used for creating the roughness maps

| Name | Abbreviation | Spatial Resolution (m) | Number of classifica- tions | Satellite Coverage Date | Reference |
|---|---|---|---|---|---|
| Global Land Cover Classification | GLCC 1000 | 1000 | 24 | 1992-1993 | USGS EROS Archive (1993) |
| MCD12Q1 MODIS/Terra+Aqua L3, v6, IGBP | MODIS 500 | 500 | 17 | 2018 | Friedl, M., Sulla-Menashe (2019) |
| C3S Global Land Cover | GLOB 300 | 300 | 38 | 2015 | European Space Agency (ESA) Climate Change Initiative (CCI) (2015) |
| CORINE land cover | CORINE 100 | 100 | 44 | 2018 | Copernicus Land Monitoring Service (2019) |





training data have been collected for all land cover classes and used as a dependent variable in a Random Forest machine learning model to predict land cover for each 20 x 20 m grid cell. Independent variables used for the land cover classification model include observations from the Sentinel-1 and Sentinel-2 sensors. LAI has been estimated for all grid cells identified as forest in the land cover classification through down-scaling of coarse resolution LAI from the MODIS sensor to the $20 \times 20$ m grid of the Sentinel-2 observations (Guzinski and Nieto, 2019).

Forest canopy heights have been estimated for all grid cells identified as forest in the land cover classification. Canopy height estimates from the ICESat-2 sensor are used as a dependent variable to train a Support Vector Regression model to predict forest canopy height for each 20 x 20 m grid cell. Observations from Sentinel-1 and Sentinel-2 are used as independent variables in the regression model. ICESat-2 provides a globally available dataset from the US National Aeronautics and Space Administration (NASA) that uses space-borne Lidar technology to estimate ground and vegetation heights. It does this with six laser beams, scanning a swath of terrain 9 km wide, with each beam having a footprint of 17 m diameter. Because of the Sentinel sensors' importance for obtaining the three layers described above, they are labelled 'Sentinel' throughout the rest of this paper.

### 3.4 Other land cover data sets

For an experienced wind resource engineer, the most accurate way to obtain a land cover dataset is by digitizing the land cover areas with the biggest impact on the flow and assign a representative $z_0$ to each area. 'Hand-digitized' maps were made with this procedure at Alaiz, Cuauhtémoc, Humansdorp, Merida and Risø (see Table 2). Another approach to obtain an accurate land cover map is by using lidar scans to estimate $h$ and applying the ORA approach (see Sect. 2.2.1). The cloud point data are described in the next section, and after obtaining $h$, the procedure described in Sect. 3.2.1 in Floors et al. (2018) was used to estimate $z_0$ and $d$ at Ryningsnäs, Sweden and Østerild. These maps are denoted as 'Lidar scans ORA 20' throughout the rest of the paper (see Table 2).

### 3.5 Elevation data

Elevation data from aerial lidar scans are available for seven of the ten sites analysed here (Table 2). The lidar scans can be used to retrieve the terrain elevation and estimate the canopy height at forested sites (Popescu et al., 2003; Floors et al., 2018).

The Swedish sites are covered by a national laser campaign conducted in 2013 (Lantmäteriet, 2016). The retrieved elevation data is available with a resolution of 0.5 first reflections $\mathrm{m^2 m^{-2}}$ and 20 m x 20 m grid spacing. Further details about the data set are given in (Floors et al., 2018). The Finland elevation model is a digital terrain model produced by the National Land Survey of Finland (Maanmittauslaitos, MML) available at a 10 m x 10 m grid. At the Danish site Østerild, lidar scans are available with a resolution of 0.5 first reflections $\mathrm{m^2 m^{-2}}$ whereas for Risø, elevation data are obtained from the 2.5 m contour lines from "Danmarks Højdemodel" (Styrelsen for Dataforsyning og Effektivisering, 2016).

The two sites Perdigão and Alaiz are particularly complex with large elevation differences and steep slopes. It is well known that the BZ flow model fails in these conditions (Bowen and Mortensen, 1996; Troen and Hansen, 2015). Nonetheless, they can be used to ensure that the developed model does not deteriorate the results for any sites. Aerial lidar scans have been





obtained over Perdigão in 2015 (Fernando et al., 2019) and over Alaiz in 2011–12 (Santos et al., 2020). Elevation data for the two Mexican sites, Cuauhtémoc and Mérida, and the site in South Africa, Humansdorp, are obtained from the Shuttle Radar

Topography Mission (JPL, 2013) at 90 m resolution.

### 3.6 Examples from Ryningsnäs

Fig. 2 illustrates the different $z_0$ representations for a $6 \times 6$ km area around the Ryningsäs mast. The GLCC database has the coarsest resolution of 1000 m, and therefore, it fails to capture any detail near the mast (a). The whole area is represented by land cover class 14, i.e. evergreen needle-leaf forest, which corresponds to $z_0 = 1.5$ m (Table A1). Likewise, the MODIS

database at 500 m resolution does not capture any detail within the selected area (b); everything is represented as evergreen needle-leaf forest with $z_0 = 1.0$ m (Table A2). Globcover, with a resolution of 300 m, represents most of the land cover around the site as evergreen needle-leaf forest with $z_0 = 1.5$ m (Table A1), and can just capture some of the lakes and open areas (c). The Corine land cover database captures more details due to its higher resolution of 100 m (d). Most of the area around the mast is classified as 'mixed forest' with $z_0 = 1.1$ m (Table A4). The lakes to the southeast of the masts and the open area to the

west of the masts are captured.

The Sentinel ORA 20 map (e) shows the result of combining the land cover map with the tree height and LAI layers according to the procedure presented in Sec. 3.3 and using the $h$ to $z_0$ and $d$ conversion from ORA (Sec. 2.2.1). The Sentinel SCADIS and Raupach maps look very similar to this and are therefore not shown. It can be seen that the map generally contains a wider range of $z_0$, ranging from 0.0002 m to over 3 m.

Finally, the $z_0$ map obtained using the ORA approach with $h$ obtained from lidar scans is shown in panel (f). Comparing the Sentinel-based tree heights (e) with tree heights derived from aerial lidar scans (f), spatially, the areas with lower and higher $h$ correspond very well. However, $z_0$ derived from Sentinel is generally lower than from the lidar scans. Because in the ORA approach, $z_0$ is specified as a fraction of $h$, this indicates a negative bias in $h$ from the Sentinel maps. To investigate this further, Fig. 3 shows the distribution of $h$ around the Ryningsnäs mast in the pixels that were classified as forest in the Sentinel maps.

The $h$ distribution from Sentinel has a mode of $h \approx 17$ m and is much more narrow than that from the lidar scans. The lidar scan distribution has the mode at $h \approx 20$ m. Furthermore the $h$ distribution from Sentinel does not have any $h > 23$ m, while the lidar scans have $h$ up to more than 33 m. This phenomenon was not only observed at Ryningsnäs, but also at the sites in Finland (Culic, 2020). We hypothesize that performing a bias correction on $h$ in the Sentinel maps will lead to better wind resource predictions. We therefore perform a test where $h$ is increased with 30%, which brings the Sentinel and lidar distributions closer

to each other.

## 4 Methodology

The WAsP model is a microscale flow model that is frequently employed for wind resource assessments (Troen and Petersen, 1989). It contains submodels for orography, roughness changes, obstacles and stability effects. In the following, we explain how the roughness and orographic submodels within WAsP are modified to better utilize the high spatial detail of $z_0$ and $d$





**Figure 2.** Roughness lengths obtained from a $6 \times 6$ km square around the Rynignsnäs site (red point) from GLCC 1000 (a), MODIS 500 (b), Globcover 300 (c), CORINE 100 (d), Sentinel ORA 20 (e) and Lidar scans ORA (f). The approach to obtain $z_0$ from a land cover class is described in the text.



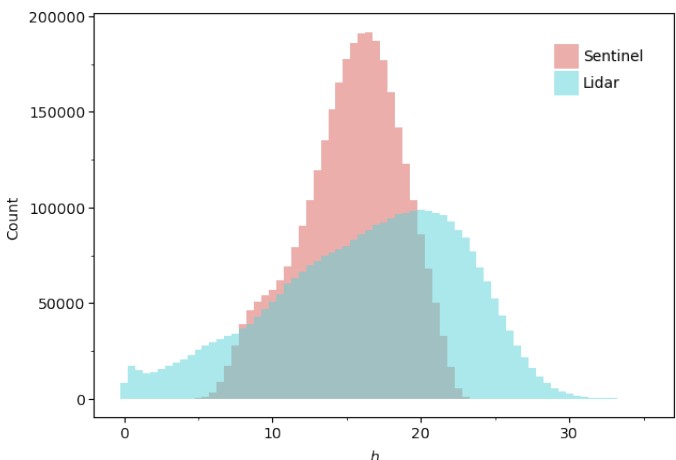

**Figure 3.** Distribution of $h$ after binning using a width of 0.5 from the aerial lidar scans and from Sentinel.

obtained from the Sentinel data set. We then describe the cross-prediction method that is applied to test the novel Sentinel data sets against more conventional input data for wind energy flow modelling. Ryningsnäs is used as an example throughout this section to illustrate our approach.

## 4.1 Model setup

The WAsP model consists of a graphical user interface (GUI) and a core model that is written in the programming language Fortran. In the following, we refer to the core model code that is directly accessed by using the Python programming language as interface. The routines are not yet available in the GUI version of WAsP but a version that has them integrated is scheduled for release in mid-2021. One of the main advantages of the WAsP core model is its speed: to calculate the AEP of a wind farm typically takes seconds. The newly implemented routines are parallelized using OpenMP in the Fortran language. Because in the WAsP core each grid point is independently calculated, the problem is easily distributed across central processing units, CPU's. To give an impression of this, the grids presented in this section consisting of 22500 points (a 6x6 km area with a resolution of 40 m) using the highest resolution maps (and thus the slowest to process), can still be calculated in less than 30 minutes on 1 node with 36 processors of DTU's supercomputer. The standard settings of the WAsP core model (corresponding to WAsP GUI version 12.6) are used here unless otherwise specified. In the following, we describe the modifications to the roughness and orographic submodules. The different parts of the model chain described in this section are shown in Fig. 4.

### 4.1.1 Forest sub-model

The ORA and Raupach models are used to obtain $z_0$ and $d$ and were implemented as described in 2.2. The SCADIS model is run without Coriolis force, because we want to obtain $z_0$ and $d$ for the logarithmic wind profile in the surface layer. The height of the model domain is specified as $4h$. The height of the first model level is specified to be $z_1 = 0.5z_{0b}$, where the background roughness length $z_{0b} = 0.3$ m. The SCADIS model is run from an initial logarithmic profile ($z_0 = z_{0b}$) with a wind speed of 10



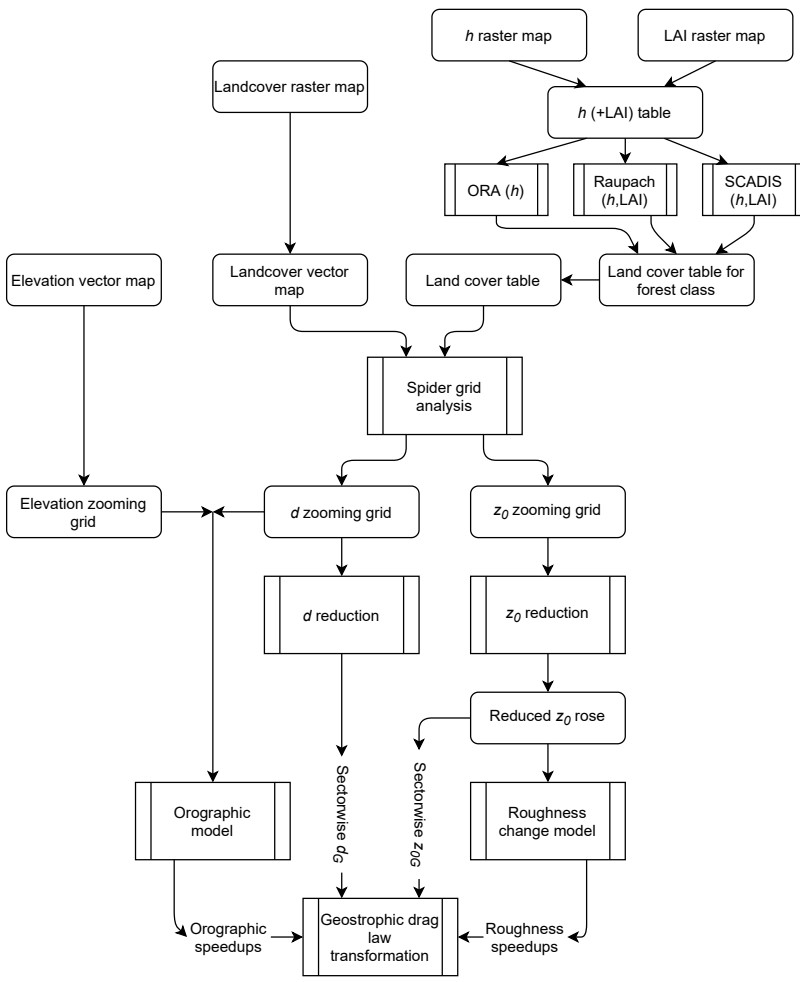

**Figure 4.** Process diagram of the sub-models in WAsP. The rounded rectangles denote data structures and the square boxes denote a method.

275   m s$^{-1}$ at the domain top. The time step in the model is set to 300 s. The model integration is terminated when the wind speed at the canopy top changes less than 0.001 m s$^{-1}$. Finally, $z_0$ and $d$ are then found by fitting $\frac{dU}{dz} = \frac{u_*}{\kappa(z-D)}$ to $U$ and $u_*$ obtained in the range from $1.5h$ to $2.5h$.

### 4.1.2   Roughness sub-model

Previous versions of WAsP use an algorithm, which finds all $z_0$ crossings between the roughness lines defined in a map and

280   a number of rays extending from the point of interest. In this study we are not only interested in $z_0$ of a land cover patch, but we also want to take into account the effect of a displacement height, $d$. This information has to be passed from the land cover contour lines to the roughness model. The roughness submodel in WAsP uses vector lines as input and therefore, the satellite based raster-maps of land cover are converted to a vector-map format.





Instead of each line in the map containing a left- and right $z_0$, the new routines expect the input of 'identifier' lines, i.e. each line contains an identifier (ID), which represents a certain land cover left and right of the line. This ID is then represented in a land cover table, which prescribes the corresponding $z_0$, $d$ and a description of that land cover class (see e.g. Tables A4). Keeping the land cover table separate from the map has several advantages, like the possibility to perform sensitivity studies with respect to $z_0$ and $d$, which has been difficult so far because one has to modify the contour map itself in an external program.

Using the approach above we can process the standard land cover maps (see Section 3.2). The Sentinel maps contain additional layers with $h$ and LAI. $h$ is discretized into bins of 5 m and the LAI into bins of 1. The result is a Sentinel land cover raster map which typically contains 10–40 different classes of forest types plus the classes specified in Table A5.

For the forest class the center of the $h$ and LAI bins are used to estimate $z_0$ and $d$ according to the three roughness models described in Section 2.2. Because the routines described in Sect 4.1.1 operate on a table and not on the contour lines in the map itself, the conversion of $h$ and LAI to $z_0$ and $d$ is fast (i.e. the speed of the computation scales with the number of entries in the land cover table).

A new routine, here referred to as a spider grid analysis, is developed to process land cover maps. It uses a polar zooming grid, similarly to the orographic flow model (Troen, 1990). The distance to the first radial segment $r_0$ in the zooming grid is defined by the user (default $r_0 = 25$ m) and each next segment has a grid spacing that is 5% larger than the previous one. The number of azimuthal bins (i.e. wind direction sectors) can also be specified by the user and by default is set to 12, i.e. using a sector width of $30°$. The first sector is always centered at the north.

For each cell in the zooming grid, the fraction of the total area $f_i$ that each of a total of $N$ land cover types in the land cover table occupies is determined and the roughness length is calculated,

$$\ln z_0 = \sum_{i=1}^{N} f_i \ln z_{0i} \tag{10}$$

The displacement height is taken into account similarly to $z_0$ and is calculated for each cell in the zooming grid as

$$d = \sum_{i=1}^{N} f_i d_i. \tag{11}$$

The zooming-grid analysis of $d$ and $z_0$ from the Sentinel ORA 20 map over Ryningsnäs is shown in Fig. 5a and 5b, respectively. For comparison with Fig. 2, we focus on a circle with a radius of 3 km. It can be seen that for the $0°$ sector for an area up to 2 km away from the mast $d \approx 10$ and $z_0 \approx 2$ m, while for the $150°$ sector at around 2 km distance there is a lake that causes lower $z_0$. Another detail visible from close inspection of Fig. 5a, is the very low values of $d$ and $z_0$ in the $150°$ sector at distances less than 100 m away from the mast. This is because of a clearing in the forest in that direction, which has important implications for the flow modelling at Ryningsäs, as will be further discussed in Sect. 5.2. We note that Ryningsnäs was also was extensively investigated by Bergström et al. (2013), who fitted a logarithmic wind profile to the measurements to obtain $z_0$ and $d$. For the $150°$ sector, $d$ was close to zero due to the clearing, while $z_0 \approx 2$ m.

Fig. 5c shows a so-called reduced roughness rose, which shows the most significant $z_0$ changes in all directions for the same area as the zooming grid analysis. This roughness rose captures the main features of the area around the site, but it also



**Figure 5.** Example of a zooming grid analysis of the displacement (a) the roughness length (b) at the Ryningsnäs site up to 3 km distance and the corresponding reduced $z_0$ rose (c) after finding the most significant $z_0$ changes.

misses some features that perhaps would have been identified 'by eye'. The large amount of clearings and forest patches in all directions clearly makes it challenging to find the most significant $z_0$ changes.

Because the effect of a roughness change on the wind speed at a certain point is distance dependent, with nearby areas having a higher impact, the $z_0$ of each area is multiplied with an exponential weighting function as described in Floors et al. (2018).

320 From these weighted values $z_{0w}$, the ones that explain most of the variance are stored for further processing (up to maximum of $n_{max}$). This is done for computational efficiency, so that equations that take into account the effect of internal boundary layers can be used. These equations are given in Sect. 8.3 in (Troen and Petersen, 1989). The output of these equations are sector-wise





speed-up factors, which are used to 'clean' the wind observations made at a certain point from microscale effects. Apart from
the speed-up factors, $z_{0w}$ is also used to compute a geostrophic (sometimes referred to as an effective or mesoscale) roughness
length $z_0G$.

Similarly to $z_0$, also $d$ has to be filtered to use it in Eq. 1 and the geostrophic drag law. Therefore, a triangular weighting
function is applied to the zooming $d$-grid. The average $d_G$ in each sector is found by taking the triangular weighted average
up to a distance $x_d = 10d_0$, with $d_0$ defined as $d$ at the first cell in the zooming grid at a distance $x_0$ from the origin. The
triangular weight $w$ is 1 at $x_0$ and $w = 0$ when $x > x_d$. Physically the reason for applying this filter is that it takes some time
for a new logarithmic profile to develop. Thus, taller trees will need longer upstream support to lead to an effective $d_G$ that is
applied in Eq. 1. The sector-wise $z_0G$ and $d_G$ can finally be used in connection with Eq. 1 and the geostrophic drag law to find
a geostrophic wind climate from an input histogram. This procedure is unaltered from the description in Chapter 8.7 in Troen
and Petersen (1989) and is therefore not further described.

### 4.1.3 Orographic submodel

The submodule for orography computes speed-ups due to elevation, and WAsP uses the Bessel-expansion on a zooming grid
(BZ) model (Troen, 1990). The input of the orographic model is a map with elevation lines, which are processed into a polar
zooming grid, with the highest resolution at the center point of the grid. The resolution of the zooming grid depends on the
radius $R$ large enough that the entire map of height contours is contained inside the circle with radius $R$. Contour lines that are
more than 20 km away from the site are ignored.

Here we are interested in the effects of $z_0$ on the flow modelling and therefore the highest quality terrain elevation map is
chosen for each of the 10 sites (see Sect. 3.1). We then study the impact of varying the land cover maps only, while keeping
the elevation map constant.

The resulting zooming grid of $d$ obtained by using Eq. 11 is added to the terrain elevation zooming grid. Using this grid,
the methodology described in Troen (1990) is used to calculate wind speed independent, sector-wise orographic speed-ups.
Similarly to the roughness speed-ups, these are used as local perturbations to the flow, which can be used to obtain a wind
climate that is representative for a large area. In a forest, the introduction of a displacement height can still cause orographic
speed-ups to be different between the Sentinel maps, although the input elevation data are the same.

### 4.2 Cross-prediction analyses

To identify the best possible baseline for flow modelling using conventional land cover data bases (see Sec. 3.2) as input, we
first perform cross-predictions using the land cover databases in combination with default land cover tables, which come with
each land cover data set. We then repeat the analyses using the revised tables suggested by Dörenkämper et al. (2020) and
Badger et al. (2015) (Appendix A). Next, the cross-predictions are performed using all other available data sets (see Table 2).

At some sites, there are multiple masts measuring at the same time (see Table 1) and the wind climate from one mast
is used to predict the wind resource at the neighbouring mast. At other sites, only one mast is available and only vertical
cross-predictions are possible, i.e. predictions of the wind distribution from one height to the next at the same mast. Only



measurements between 20 and 200 m are used to avoid using data measured within the roughness sublayer, where the WAsP model does not apply. In total, there are 1538 possible cross-predictions for evaluation of the different roughness maps.

Observed wind climates are generated for each height and mast from the 10–min. time series of wind speed $U$ and wind direction $\theta$, which are discretizised into histograms using a bin width of $1\,\mathrm{m\,s^{-1}}$ and $30\,^\circ$, respectively.

The power per unit area swept by wind turbine blades, $P$ (also referred to as the power density), is given by

$$P = 0.5\rho U^3, \tag{12}$$

where $\rho$ is the air density. In WAsP the wind distribution is described by sector-wise Weibull distributions and therefore we can conveniently find $P$ from the Weibull parameters $A$ and $k$ for $N_d$ sectors as

$$P = \rho \sum_{i=1}^{N_d} \phi_i A_i^3 \Gamma(1 + 3/k_i) \tag{13}$$

where $\phi$ is the sector-frequency and $\rho$ is calculated according to the methods described in Floors and Nielsen (2019). Throughout this paper we choose to evaluate relative errors of power density,

$$\varepsilon_P = 100\left(\frac{P_{\mathrm{mod}}}{P_{\mathrm{obs}}} - 1\right) \tag{14}$$

where $P$ is obtained from the modelled (mod) or observed (obs) Weibull distributions. The statistics reported throughout the rest of the paper are the bias, $\overline{\varepsilon_P}$, and the root-mean-square error (RMSE), $\sqrt{\overline{\varepsilon_P^2}}$, where the overbar denotes a mean. We also
include relative errors in wind speed $U$, which we obtain as described above but instead of Eq. 13, $U$ is computed from the Weibull distributions as,

$$U = \sum_{i=1}^{N_d} \phi_i A_i \Gamma(1 + 1/k_i) \tag{15}$$

As an example, we use the observed wind speed histogram at the Ryningsnäs mast at 80 m and predict the wind resource at 80 m in the same 6x6 km area as in the previous section, using a raster with a resolution of 40 m.

In Fig. 6a, the elevation is shown; the elevation changes are modest, but there is a hill to the west. In Fig. 6b, the displacement height in each point is shown. For clarity, we only focus on $d$ for the most frequently observed westerly sector. In the center of the map the clearing southeast of the mast position is visible, resulting in lower displacement heights. In addition, displacement heights of more than 10 m are visible further from the mast to the east and southeast.

The result of combined elevation, displacement and roughness description on the emergent power density in the area is
shown in Fig. 6c. The emergent power density is calculated from the frequency weighted sector-wise $A$ and $k$ parameters in each point (Eq. 13). The higher power density due to the flow speed-ups over the hill in the west are clearly visible. Because the changes in power density due to the introduction of a displacement height are difficult to discern, Fig 6d shows the difference in percent between $P$ with and without using a displacement height. Using $d$ causes a decrease in power density between 0 and 12%. The area with highest reduction in power density in the center indeed corresponds closely to the area with the highest
displacement heights in Fig 6b.

**Figure 6.** Map with the terrain elevation (a), the displacement height (b), the power density (c) and the relative difference in power density compared to a terrain map where $d = 0$ (d) at Ryningsnäs for westerly winds.





## 5 Results

In the following, we show results of the cross-predictions. We first examine the results per site and then we aggregate results for all the 10 sites investigated. Because the WAsP model is known to perform poorly in complex terrain with steep slopes, we exclude sites with a RIX-number higher than 1% in the aggregation (see Table 1). Applying this filter leaves 1082 cross-390    prediction for the GLCC 1000 and GLOB 300 databases (global coverage) and 1034 for the CORINE 100 (only Europe) database.

### 5.1 Cross-predictions at wind energy sites

The RMS of $\varepsilon_P$ per site is shown in Fig. 7. The complexity of the Perdigão and Alaiz sites clearly leads to very large errors, which highlights that higher fidelity flow models that can deal with flow separation should be used at these sites and they will
not be included for further analysis. Other sites where we find rather large errors include Finland and Ryningsnäs. Both are forested sites with complex vegetation patterns. This indicates a need for better flow modelling at such in-homogeneous sites. Humansdorp and Cuauhtémoc, in contrast, are characterized by a relatively simple terrain, which results in a low RMS of $\varepsilon_P$. The cross-predictions performed using the novel Sentinel maps as input mostly lead to lower RMS of $\varepsilon_P$ than those using the standard land cover databases. The three forest roughness models lead to very similar results at all 10 sites.

If we consider only the cross-predictions for sites where aerial lidar scans are available (i.e. Ryningsnäs, Sweden, Finland, and Østerild), we see that the RMSE of $\varepsilon_P \approx 10.3\%$ for the aerial lidar scans (Lidar scans ORA 20) and $11.4\%$ for the Sentinel ORA 20 maps, respectively. At Ryningsnäs, Finland, and Østerild the aerial lidar scans yield lower RMS of $\varepsilon_P$ than the Sentinel maps, whereas at the Swedish site results are more comparable. This will be further discussed in Sec. 5.2. When we consider the sites with hand-digitized maps only (i.e. Humansdorp, Alaiz, Merida, Cuauhtémoc, Risø), the RMS of $\varepsilon_P$ is 23.8% for the
hand-digitized map, 27.1% for the Sentinel ORA 20 map and 31.9% for the GLOB 300 map. Thus, improvements in RMS of $\varepsilon_P$ when using Sentinel maps compared to hand-digitized maps cannot not be shown.

### 5.2 Effect of tree height on cross-predictions

The replacement of $h$ with $1.3h$ in the ORA 20 roughness model leads to a slightly lower RMS of $\varepsilon_P$ for most of the sites (Fig. 8). This is due to the higher $z_0$ and $d$ that directly results from the higher $h$. However, the Ryningsnäs site shows a RMS
of $\varepsilon_P$ that is twice as low compared to simulations using $h$ from the original Sentinel maps. This large improvement in RMSE is due to the location of the Ryningsnäs mast very close to the forest edge. For the westerly sector, $d$ is nearly doubled, which has large implications for the predicted power density (see Fig. 6). This indicates that for masts located very close to forest edges, sensitivity studies are still required and the Sentinel maps should be carefully validated against measured tree heights. Multiplying $h$ with 1.3 decreased the RMS of $\varepsilon_P$ of simulations with the Sentinel Raupach and SCADIS maps similarly to the
Sentinel ORA maps and these are therefore not shown.





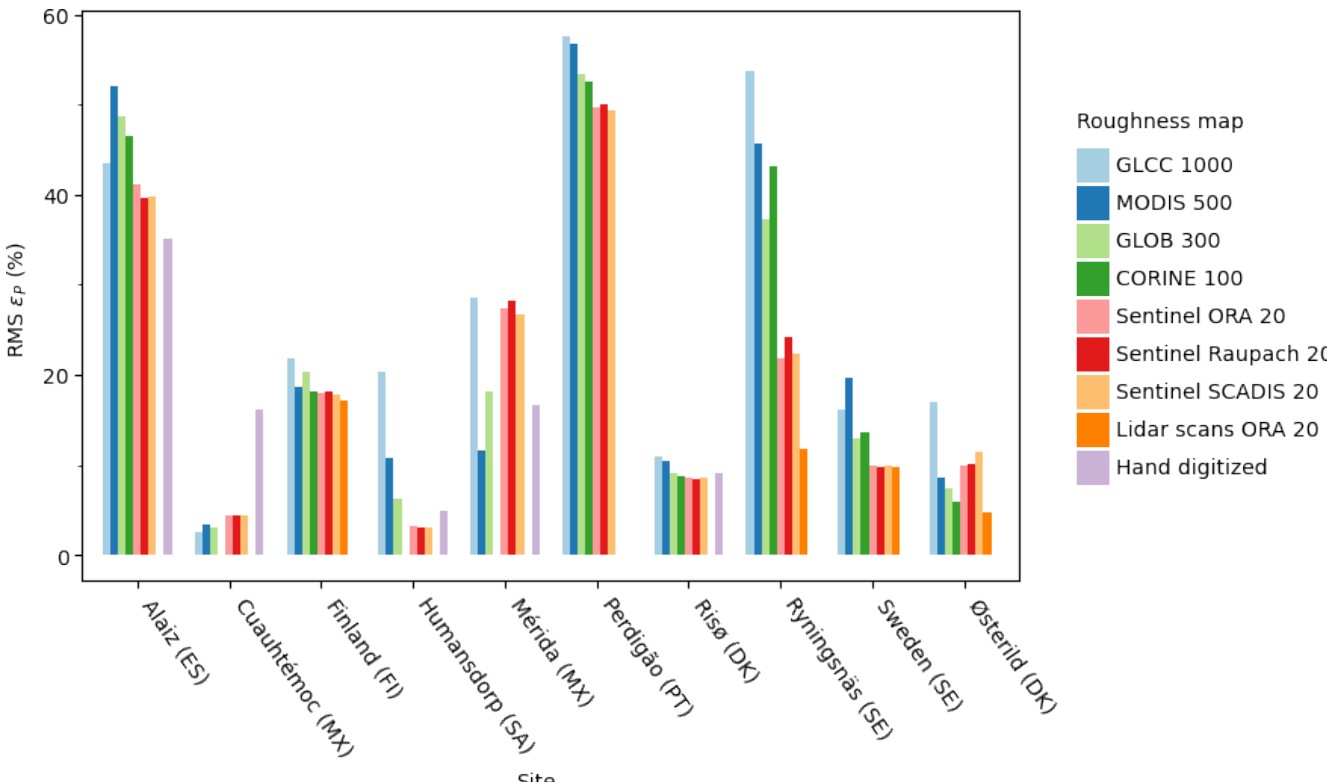

**Figure 7.** RMS of $\varepsilon_P$ at all different sites, including those in highly complex terrain.

## 5.3 Aggregated results of cross-predictions

We can now aggregate results from the cross-predictions for the eight sites with low complexity (RIX<1%) to obtain the average performance of each set of input data in connection with flow modelling in WAsP.

The RMSE in power density for the cross-predictions using standard land cover databases with original and revised rough-ness translation tables is shown in Fig. 3. The GLCC 1000 generally leads to the highest errors, both when the original and revised land cover tables are used. The lowest RMSE of all is achieved with the GLOB 300 maps in combination with the revised land cover table. The RMSE is reduced for all land cover databases when the revised land cover tables are used instead of the original ones.

We can now compare results generated with the standard land cover databases with results generated with the novel Sentinel data layers as input. Fig 10 shows the prediction errors when the WAsP model is run with six different inputs: the three global standard land cover data bases with revised roughness translation tables plus three types of Sentinel maps based on the three different forest roughness models described in Section 2.2. The CORINE 100 database is not available at all sites and is therefore not shown. Similar result were obtained when selecting only European sites (not shown).

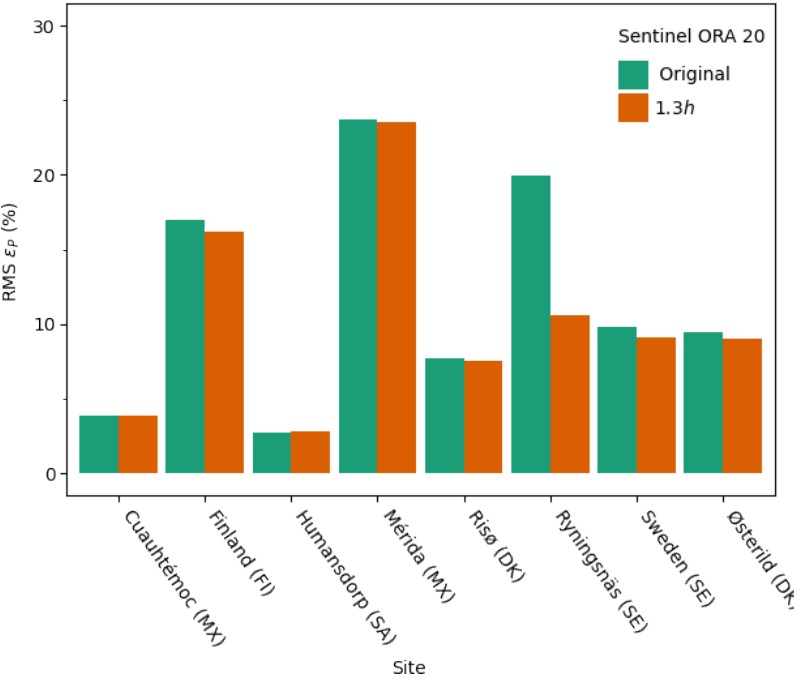

**Figure 8.** RMS $\varepsilon_P$ at simple sites for the model runs with Sentinel ORA 20 maps (Original) and those where the tree height was increased with 30% .

All maps that share a more advanced forest model, yield lower RMSE (RMS of $\varepsilon_P \approx 11.7$) than the standard land cover

databases. Table 4 shows that also the mean bias is lower in the Sentinel maps than in the standard land cover databases. Similarly, the errors in wind speed $\varepsilon_U$ are also smaller for the Sentinel maps model runs. The mean bias in $\varepsilon_U$ becomes slightly negative, but is generally close to zero for all WAsP model runs.

When running WAsP it is often more instructive to evaluate horizontal cross-predictions, because the roughness rose will then be different for two points. To investigate this, the data was additionally filtered to include only horizontal cross-predictions.

However, this did not change the large differences in $\varepsilon_P$ and $\varepsilon_U$ between the model simulations based on land cover databases and Sentinel maps. Also the effect of measuring height was investigated by only including cross-predictions between 50 and 200 m, but also this did not change the general picture that emerges from Table 4.

## 6    Discussion

For the first time, forest parameters retrieved from the Sentinel and ICESat-2 satellites have been used for wind energy flow

modelling. Our examples demonstrate that the variability of the land cover within forests (e.g. vegetation with different height and density, clearings, lakes) is resolved far better by these novel products than standard land cover products with pan-European or global coverage. The high level of spatial detail in these satellite-based data layers is almost comparable to products derived



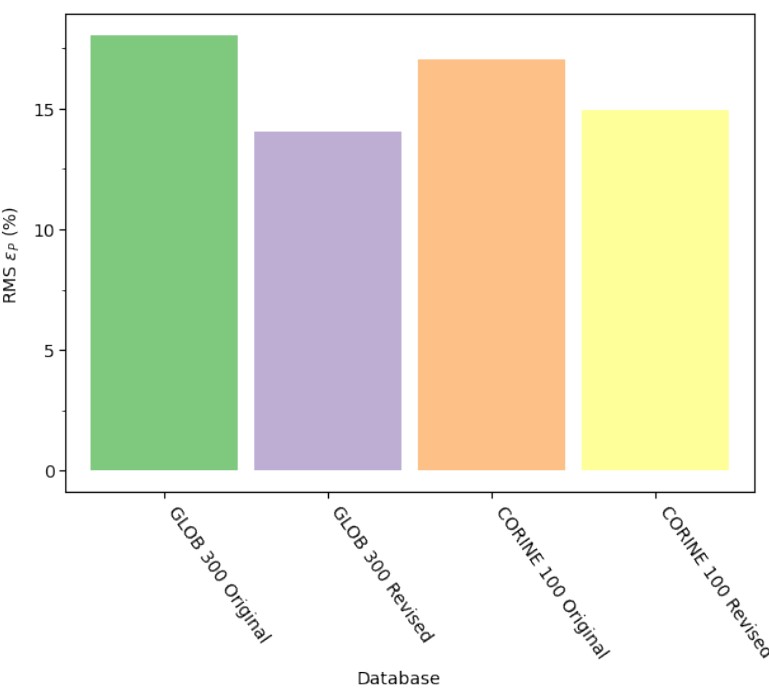

**Figure 9.** RMS error in power density ($\varepsilon_P$, in %) at the low ($< 1\%$) RIX sites using the original and revised land cover databases (see text and Tables A4 and A3 in the Appendix). MODIS 500 and GLCC 1000 have only one recommended $z_0$ table and are therefore not shown.

**Table 4.** Errors when modelling at all sites with low complexity (RIX$< 1\%$)

| Roughness map | RMS $\varepsilon_P$ (%) | Mean bias $\varepsilon_P$ (%) | RMS $\varepsilon_U$ (%) | Mean bias $\varepsilon_U$ (%) |
|---|---|---|---|---|
| GLCC 1000 | 19.1 | 2.5 | 6.5 | 0.3 |
| MODIS 500 | 19.8 | 2.7 | 6.6 | 0.3 |
| GLOB 300 | 14.5 | 1.9 | 5.1 | 0.1 |
| Sentinel ORA 20 | 11.6 | 0.6 | 4.6 | -0.2 |
| Sentinel Raupach 20 | 11.7 | 0.1 | 4.7 | -0.4 |
| Sentinel SCADIS 20 | 11.7 | 0.6 | 4.6 | -0.2 |



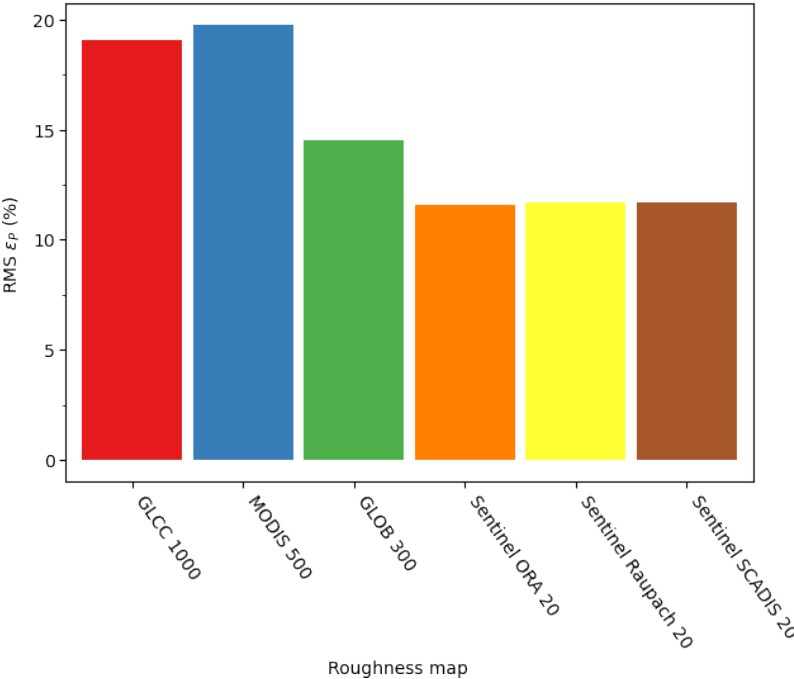

**Figure 10.** RMS of $\varepsilon_P$ at all sites with low complexity (RIX< 1%).

from aerial lidar scans. This is promising in connection with wind energy flow modelling because the satellite-based data layers can be produced for any site in the world and updated frequently. The cost of having the satellite-based forest data layers is far
lower than the price of dedicated airborne lidar campaigns thanks to the open access to satellite data archives by Copernicus and NASA.

The observed bias in $h$ in the ICESat-2/Sentinel product compared to the lidar scans at Ryningsnäs may partly be due to the different sensing and retrieval methods applied and partly due to temporal differences of the two tree height products. For example, the standard canopy height product from ICESat-2 (ATL08) records heights over a $17 \times 100$ m along-track transect,
giving a mismatch with resolution of the lidar scans. Additionally, the tree-growth in temperate forests can be up to a few m per year. Therefore, it is important to use the most updated $h$ parameterization for the modelling of wind resources. Our cross-prediction analyses reveal that adding 30% to tree heights retrieved from Sentinel can improve the accuracy of wind power predictions. Further research is needed to fully understand and improve the absolute accuracy of tree heights retrieved from satellite observations. Likewise, the absolute accuracy of forest densities expressed through the LAI should be thoroughly
tested in future work.

Our analyses show minor differences in RMSE when cross-predictions are performed using Sentinel vs aerial lidar scans to estimate $d$ and $z_0$. These findings are promising in the light of the lower cost and the global coverage of the satellite-based data layers. Cross-predictions performed with a manual assessment of $z_0$ lead to a lower RMSE at 2 of the 5 sites in our cross-





predictions than any of the automatic assessments based of satellite observations or aerial lidar scans. However, automated
procedures can speed up wind resource assessments and remove the subjective judgement of a siting engineer.

Our results based on standard land cover databases improve when the land cover to roughness translation tables are modified
specifically for land cover classes related to forests. For wind energy projects within forests, it is, therefore, advisable to use
the newly recommended translation tables A3 and A4 shown in Appendix A rather than the translation tables readily available
through software tools for flow modelling.

Although the differences between the three forest models are small in this study, we find that the Raupach and Scadis model
perform slightly better than the ORA model when a higher $h$ is used. This indicates that some improvements can be made by
including LAI. The LAI obtained from the Sentinel maps is based on very coarse resolution input data. Therefore, the SCADIS
forest model is likely too advanced for the data used in this study due to the large uncertainty in the input data and the absence
of any information about the canopy density profile. More detailed studies at a higher number of sites, preferably with observed
$h$ and LAD profiles, are needed to validate this result further.

Using a higher $z_0$ than derived from the maps presented in this paper generally gives better results for the forested sites.
This is thought to be related to roughness length aggregation, i.e. finding $z_0$ for a larger area that correctly prescribes the
momentum flux in surface-layer similarity in connection with Eq. 1. The simple logarithmic averaging applied in Eq. 10
is known to be flawed in heterogeneous conditions. For example, Taylor (1987) suggested accounting for sub-grid variance
of $z_0$ when calculating an effective roughness length. Vihma and Savijärvi (1991) compared model approaches for several
landscape configurations in Finland and found that $z_{0eff}$ was always higher than a simple logarithmic average. This was also
confirmed by Hasager and Jensen (1999), who found that the difference between the logarithmic average and $z_{0eff}$ was higher
for landscapes with small patches and with a half-to-half mix of rough and smooth patches. For small patches, one can argue
that the form drag of the forest will always lead to a much higher $z_{0eff}$ than implied through logarithmic averaging. However,
an analysis of the effect of $z_0$ aggregation was not attempted in this study, but it's impacts are likely large for heterogeneous
sites like Østerild, Finland and Ryningsnäs. Bottema et al. (1998) reviewed a range of $z_0$ aggregation methods, which could be
investigated in future work.

In this study, cross-prediction is used without any manual interference. For an actual wind energy project, much smaller
errors would be achieved using the WAsP model because the site engineer usually has access to measurements at several
heights, which can be used to fit the model nearly perfectly to the measurements. However, it is still important to create a
$z_0$ map that is as accurate as possible because also stability can cause notable variations in the wind profile. The effect of
atmospheric stability is beyond the scope of this study, and the default settings in WAsP are used, but it is acknowledged that
there can be deviations at some sites.

We note that wind engineers should be careful when using databases such as SRTM in combination with a displacement
length. Because the SRTM product shows the height of the surface, it includes the tree height. If this is not taken into account,
the turbine or mast might be placed at too high elevation, and there is a risk of double-counting of the effects of the forest.



# 7 Conclusions

We have tested a novel satellite-based product for land surface parameterization in forested areas and quantified the effect of using this product for wind resource modelling. The novel satellite-based product is based on observations from Sentinel-1, -2 and ICESat-2 and it contains collocated layers of land cover, tree heights and LAI at a 20-m spatial resolution. These maps are converted to maps of roughness lengths and displacement heights using three different forest modules of varying complexity. The simplest way to convert the tree height to $z_0$ and $d$ is by multiplying with a constant (e.g. 1/10). Secondly, a physically more advanced approach, which also considers the effect of LAI, is implemented. The third module is a 1D version of the SCADIS model and it takes the effects of varying LAI and canopy density profiles into account. All three forest modules are used in the WAsP model, which is frequently used for prediction the wind resources from wind measurements.

We find that the high complexity of forested landscapes is poorly resolved by global and pan-European land cover databases such as GLCC, MODIS, CORINE and Globcover. The novel satellite-based product leads to more detailed maps of $z_0$ and $d$, which are spatially comparable to aerial lidar scans or hand-digitized maps. The $z_0$ retrieved from satellites has a negative bias compared to the aerial lidar scans and further research is needed to refine the product.

Cross-predictions have been performed at 10 sites with tall masts to evaluate the effect of using different input data sets in connection with flow modeling in WAsP. The novel maps from Sentinel lead to a reduction of the RMS of relative errors in power density at most sites and on average by $\approx 3\%$ points compared to the best performing roughness map obtained from a coarse-resolution land cover database. This is even after the roughness lengths for specific land cover categories in the coarse-resolution products are improved. Differences between the three forest modules are minor, showing that the sensitivity of the WAsP model to different approaches to obtain $z_0$ and $d$ is low.

The RMS of relative errors in power density found for the Sentinel maps (11.4%) are comparable to those obtained from aerial lidar scans (10.3%). This finding is very promising because the novel satellite-based maps of $z_0$ and $d$ can be generated at a lower cost and a higher temporal resolution than aerial lidar campaigns. Processing of the satellite-based maps is fully automated. For sites that show a potential for wind power projects, the new routines and products could replace current practices of land cover analysis, which is time-consuming and plagued by subjective assessments.

## Appendix A

The land cover tables for the land cover databases that were investigated in this paper are supplied in Tables A1-A5. The references to the land cover databases can be found in Sec. 3.2.

*Code and data availability.* The numerical results are generated with proprietary software.



**Table A1.** land cover table used for the GLCC land cover database.

|  | Description | Ori. z0 | Rev. z0 |
|---|---|---|---|
| 1 | Urban and Built-Up Land | 0.400 | 1.000 |
| 2 | Dryland Cropland and Pasture | 0.100 | 0.100 |
| 3 | Irrigated Cropland and Pasture | 0.100 | 0.050 |
| 4 | Mixed Dryland/Irrigated Cropland and Pasture | 0.100 | 0.100 |
| 5 | Cropland/Grassland Mosaic | 0.070 | 0.070 |
| 6 | Cropland/Woodland Mosaic | 0.150 | 0.150 |
| 7 | Grassland | 0.050 | 0.030 |
| 8 | Shrubland | 0.070 | 0.200 |
| 9 | Mixed Shrubland/Grassland | 0.060 | 0.100 |
| 10 | Savanna | 0.070 | 0.070 |
| 11 | Deciduous Broadleaf Forest | 0.400 | 1.500 |
| 12 | Deciduous Needleleaf Forest | 0.400 | 1.500 |
| 13 | Evergreen Broadleaf Forest | 0.500 | 1.500 |
| 14 | Evergreen Needleleaf Forest | 0.500 | 1.500 |
| 15 | Mixed Forest | 0.400 | 1.500 |
| 16 | Water Bodies | 0.000 | 0.000 |
| 17 | Herbaceous Wetland | 0.030 | 0.030 |
| 18 | Wooded Wetland | 0.100 | 0.400 |
| 19 | Barren or Sparsely Vegetated | 0.020 | 0.010 |
| 20 | Herbaceous Tundra | 0.050 | 0.030 |
| 21 | Wooded Tundra | 0.150 | 0.300 |
| 22 | Mixed Tundra | 0.100 | 0.100 |
| 23 | Bare Ground Tundra | 0.030 | 0.010 |
| 24 | Snow or Ice | 0.001 | 0.003 |

**Table A2.** land cover table used for the MODIS based maps.

|  | Description | z0 |
|---|---|---|
| 0 | Water | 0.000 |
| 1 | Evergreen Needle leaf Forest | 1.000 |
| 2 | Evergreen Broadleaf Forest | 1.000 |
| 3 | Deciduous Needle leaf Forest | 1.000 |
| 4 | Deciduous Broadleaf Forest | 1.000 |
| 5 | Mixed Forests | 1.000 |
| 6 | Closed Shrublands | 0.050 |
| 7 | Open Shrublands | 0.060 |
| 8 | Woody Savannas | 0.050 |
| 9 | Savannas | 0.150 |
| 10 | Grasslands | 0.120 |
| 11 | Permanent Wetland | 0.300 |
| 12 | Croplands | 0.150 |
| 13 | Urban and Built-Up | 0.800 |
| 14 | Cropland/Natural Vegetation Mosaic | 0.140 |
| 15 | Snow and Ice | 0.001 |
| 16 | Barren or Sparsely Vegetated | 0.010 |



**Table A3.** land cover table used for the Globcover/ESA-CCI land cover database.

|     | Description | Ori. z0 | Rev. z0 |
| --- | --- | --- | --- |
| 0 | No data | 0.000 | 0.000 |
| 10 | Cropland, rainfed | 0.100 | 0.100 |
| 11 | Cropland rainfed, Herbaceous cover | 0.100 | 0.100 |
| 12 | Cropland rainfed, Tree or shrub cover | 0.200 | 0.200 |
| 20 | Cropland, irrigated or post-flooding | 0.070 | 0.050 |
| 30 | Mosaic cropland (>50%) / natural vegetation (t... | 0.070 | 0.200 |
| 40 | Mosaic natural vegetation (tree, shrub, herbac... | 0.500 | 0.300 |
| 50 | Tree cover, broadleaved, evergreen, closed to ... | 0.400 | 1.500 |
| 60 | Tree cover, broadleaved, deciduous, closed to ... | 0.400 | 1.000 |
| 61 | Tree cover, broadleaved, deciduous, closed (>40%) | 0.400 | 1.000 |
| 62 | Tree cover, broadleaved, deciduous, open (15-40%) | 0.400 | 0.800 |
| 70 | Tree cover, needleleaved, evergreen, closed to... | 0.500 | 1.500 |
| 71 | Tree cover, needleleaved, evergreen, closed (>... | 0.500 | 1.500 |
| 72 | Tree cover, needleleaved, evergreen, open (15-... | 0.500 | 1.500 |
| 80 | Tree cover, needleleaved, deciduous, closed to... | 0.500 | 1.200 |
| 81 | Tree cover, needleleaved, deciduous, closed (>... | 0.500 | 1.200 |
| 82 | Tree cover, needleleaved, deciduous, open (15-... | 0.500 | 1.200 |
| 90 | Tree cover, mixed leaf type (broadleaved and n... | 0.400 | 1.500 |
| 100 | Mosaic tree and shrub (>50%) / herbaceous cove... | 0.400 | 0.200 |
| 110 | Mosaic herbaceous cover (>50%) / tree and shru... | 0.070 | 0.100 |
| 120 | Shrubland | 0.070 | 0.100 |
| 121 | Shrubland evergreen | 0.070 | 0.200 |
| 122 | Shrubland deciduous | 0.070 | 0.200 |
| 130 | Grassland | 0.070 | 0.030 |
| 140 | Lichens and mosses | 0.050 | 0.010 |
| 150 | Sparse vegetation (tree, shrub, herbaceous cov... | 0.070 | 0.050 |
| 151 | Sparse tree (<15%) | 0.070 | 0.050 |
| 152 | Sparse shrub (<15%) | 0.070 | 0.050 |
| 153 | Sparse herbaceous cover (<15%) | 0.070 | 0.050 |
| 160 | Tree cover, flooded, fresh or brakish water | 0.100 | 0.800 |
| 170 | Tree cover, flooded, saline water | 0.100 | 0.600 |
| 180 | Shrub or herbaceous cover, flooded, fresh/sali... | 0.400 | 0.100 |
| 190 | Urban areas | 0.400 | 1.000 |
| 200 | Bare areas | 0.020 | 0.005 |
| 201 | Consolidated bare areas | 0.020 | 0.005 |
| 202 | Unconsolidated bare areas | 0.020 | 0.005 |
| 210 | Water bodies | 0.000 | 0.000 |
| 220 | Permanent snow and ice | 0.001 | 0.003 |





**Table A4.** land cover table used for the CORINE land cover database.

| | Description | Ori. z0 | Rev. z0 |
|---|---|---|---|
| 0 | No data | 0.0000 | 0.000 |
| 48 | No data | 0.0000 | 0.000 |
| 255 | No data | 0.0000 | 0.000 |
| 1 | Continuous urban fabric | 0.5000 | 1.000 |
| 2 | Discontinuous urban fabric | 0.4000 | 0.800 |
| 3 | Industrial or commercial units | 0.7000 | 0.700 |
| 4 | Road and rail networks and associated land | 0.1000 | 0.100 |
| 5 | Port areas | 0.5000 | 0.500 |
| 6 | Airports | 0.0300 | 0.010 |
| 7 | Mineral extraction sites | 0.1000 | 0.050 |
| 8 | Dump sites | 0.1000 | 0.050 |
| 9 | Construction sites | 0.3000 | 0.300 |
| 10 | Green urban areas | 0.4000 | 0.800 |
| 11 | Sport and leisure facilities | 0.5000 | 0.200 |
| 12 | Non-irrigated arable land | 0.0560 | 0.050 |
| 13 | Permanently irrigated land | 0.0560 | 0.030 |
| 14 | Rice fields | 0.0184 | 0.030 |
| 15 | Vineyards | 0.3000 | 0.300 |
| 16 | Fruit trees and berry plantations | 0.4000 | 0.400 |
| 17 | Olive groves | 0.4000 | 0.400 |
| 18 | Pastures | 0.0360 | 0.030 |
| 19 | Annual crops associated with permanent crops | 0.0560 | 0.100 |
| 20 | Complex cultivation patterns | 0.0560 | 0.150 |
| 21 | Land principally occupied by agriculture with ... | 0.0560 | 0.200 |
| 22 | Agro-forestry areas | 0.5000 | 0.500 |
| 23 | Broad-leaved forest | 0.5000 | 1.000 |
| 24 | Coniferous forest | 0.5000 | 1.200 |
| 25 | Mixed forest | 0.5000 | 1.100 |
| 26 | Natural grasslands | 0.0560 | 0.030 |
| 27 | Moors and heathland | 0.0600 | 0.050 |
| 28 | Sclerophyllous vegetation | 0.0560 | 0.070 |
| 29 | Transitional woodland-shrub | 0.4000 | 0.400 |
| 30 | Beaches - dunes - sands | 0.0100 | 0.003 |
| 31 | Bare rocks | 0.0500 | 0.050 |
| 32 | Sparsely vegetated areas | 0.2000 | 0.030 |
| 33 | Burnt areas | 0.2000 | 0.200 |
| 34 | Glaciers and perpetual snow | 0.2000 | 0.005 |
| 35 | Inland marshes | 0.0500 | 0.050 |
| 36 | Peat bogs | 0.0184 | 0.030 |
| 37 | Salt marshes | 0.0348 | 0.020 |
| 38 | Salines | 0.0300 | 0.005 |
| 39 | Intertidal flats | 0.0005 | 0.000 |
| 40 | Water courses | 0.0000 | 0.000 |
| 41 | Water bodies | 0.0000 | 0.000 |
| 42 | Coastal lagoons | 0.0000 | 0.000 |
| 43 | Estuaries | 0.0000 | 0.000 |
| 44 | Sea and ocean | 0.0000 | 0.000 |



**Table A5.** land cover table used for the Sentinel based maps.

|   | Description | z0 |
|---|---|---|
| 0 | Non-forest (cropland, grassland, other) | 0.03 |
| 1 | Forest | ORA, Raupach or SCADIS model |
| 2 | Water bodies | 0.00 |
| 3 | Urban/Built up | 1.00 |
| 4 | Open forest | 0.40 |

*Author contributions.* RF generated the results, contributed to the WAsP model code and drafted the manuscript. MB coordinated the work and revised the manuscript. IT developed most of the methods and code in the WAsP model and contributed to the preparation of the manuscript. KG created the Sentinel data layers and wrote a section of the manuscript. FP helped with data-processing and preparation of the manuscript.

*Competing interests.* The high-resolution forest data layers generated from Sentinel are sold by DHI GRAS A/S, where KG is employed.
The WAsP software is maintained and sold by DTU Wind Energy, where RF and MB are employed.

*Data availability.* Sample data packages containing the Sentinel forest data layers for selected sites are available at https://help.emd.dk/ mediawiki/index.php?title=Innowind_Premium_Data_Layers. The wind measurements from the mast in South Africa (WM08) are available at http://wasadata.csir.co.za/wasa1/WASAData. Data from the two Mexican masts (M02 and M03) is available at https://aems.ineel.mx/ aemdata/MemberPages/Download.aspx?lang=EN.

*Acknowledgements.* This work has received funding from the H2020 e-shape project (grant agreement 820852), from Innovation Fund Denmark through the InnoWind project (6172-00004B), and from the Ministry of Foreign Affairs of Denmark administered by Danida Fellowship Centre through the 'multi-scale and model-chain evaluation of wind atlases' (MEWA) project (17-M01-DTU). We acknowledge the providers of global and pan-European land cover data sets: US Geological Survey for GLCC and MODIS, ESA Climate Change Initiative and in particular its Land Cover project as the source of the CCI-LC database, and ESA Land Monitoring Service for CORINE. ESA and
the European Commission are acknowledged for Copernicus Sentinel data. NASA are acknowledged for ICESat-2 data, and in particular for regular consultation through the Early Adopter Program. The Ingeborg and Leo Dannin Prize for Scientific Research (2013) is acknowledged for the aerial lidar campaign over Perdigão. Ebba Dellwik from DTU is thanked for preparation of aerial lidar data and Andreas Bechmann for his review of the paper. We acknowledge the Wind Atlas for South Africa project for the wind observations near Humansdorp (CSIR, 2020) and the Mexican Wind Atlas project for observations at the Mexican sites (INEEL, 2020). Technicians of DTU are acknowledged for
the maintenance of the Risø and Østerild masts.



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
