# Peer review of "Satellite-based estimation of roughness lengths and displacement heights for wind resource modelling"

_Wind Energy Science, 2021_

## Referee Comment (RC1)

**Comments on the paper from Floors et al. 'Satellite-based estimation of roughness lengths and displacement heights for wind-resource modelling'**

**General comments**

The presented paper is concerned with improving wind-energy potential prediction in forested areas or inhomogeneous terrain, where obstacles influence the local wind profile often unpredictably. The authors' approach is to use satellite-based spatial data to retrieve the roughness length $z_0$ and and displacement length $d$, instead of commonly used tables or even manual assigned parameters, e.g. derived from in-situ measurements or 'hand-digitising'. The authors compare their novel satellite-based flow modelling results to predictions based on global land cover maps, lidar scans and manual digitisation. Their goal is to increase the accuracy of wind cross-prediction and reduce uncertainties that come with the traditional methods in cross-prediction in heterogeneous terrain. Their results show a slightly better cross-prediction. The results depend on the complexity of the terrain though. The significance of the paper, however, is not only based on the potential increase of accuracy in cross-prediction, but that the satellite-based approach is using an available and suitable source for (seasonable) land coverage.

The authors present a lot of data bases, methods and models aside with an exemplary data evaluation. This can be a lot at times and needs to be carefully balanced to keep it to necessary information for the reader.

In general the paper is well structured and their methods are well presented. However, there are some (minor) issues with the current state of the manuscript. All issues are listed in the 'Specific comments' section.

**Specific comments**

l. 91: Please introduce $h$ properly as the canopy height.

l. 130: Here you discuss the plot shown in Fig. 1, especially around the shown LAI around 1. This is where the models differentiate from each other. However, the legend of the plot is overlapping with a large portion of the plot that is being discussed. This can be solved easily by moving the legend in a way that it does not interfere with the graphs. Alternatively increase the y-axis limits and lift the legend up to around 1.0-1.3 on the y-axis.

l. 134: You mention the differences for different $\alpha$ and $\beta$ values in the respective models. You should state in a small statement which one you will use in the study and if you stick to the temperate pine forest. Also add the information where these kind of forest parameters shown in line 134 apply to.

l. 256: 'Ryningsnäs is used as an example throughout this section to illustrate our approach'. This is a sentence one would expect at section 3.6. where you introduce the example 'Ryningsnäs'. You could even move Sec. 3.6 into Sec. 4.

l. 296ff: You introduce the spider-grid analysis or zooming-grid analysis. Can you elaborate why this is used instead of an orthogonal grid as in the later predictions?

l. 365: Please introduce $\Gamma$ from Eq. 13.

l. 398: [...] 'mostly' lead to lower RMS ... Maybe use more like 'in half of the cases' (6/10). And even then only by a small margin. It would be more representative of the figure.

l. 404: Are those $\epsilon_p$ values averages for all the sites combined, you mention in line 404? Since you mention different sites but only one RMS and method.

l. 405: Can you elaborate why those improvements can not be shown? Or how to understand this conclusion. Do you mean the data does not show this? Because for the Cuauhtémoc site the hand-digitised results show a higher RMS. Maybe clear this paragraph up.

l. 411f: Please concretise 'westerly sector', e.g. using the easting that can be used in Fig. 6., as it additional seems that the values for $d$ in average double in the easterly part of the plot (more yellow and green).

l. 413: This is unfortunate as you try to make a case for the Sentinel satellite data, especially a tool for large area with no mast data to validate.

l. 420: Why does Fig. 9 exist? It shows four bars which height difference is not really quantifiable from the graph. The information it is supposed to deliver could be added to Table 4 instead by adding an extra line for the original data base. Or add a column that deals with it.

l 453f: This should be moved to the conclusions part of the study.

**Technical corrections**

l. 165: You refer to Table 3 long before you mention Table 2 in line 206. I would prefer it, if you swap the labelling to avoid confusion.

l. 420: Wrong cross-reference. I think you mean Fig. 9 not Fig. 3.

l. 451: Please spell 'meters' out in this context.

---

## Referee Comment (RC2)

[referee-annotated manuscript omitted]

---

## Author Comment (AC2)

[revised manuscript text omitted]

---

## Author Response (AR1)

**Reviewer 1**

Comments on the paper from Floors et al.'Satellite-based estimation of roughness lengths and displacement heights for wind-resource modelling'

**General comments**

The presented paper is concerned with improving wind-energy potential prediction in forested areas or inhomogeneous terrain, where obstacles influence the local wind profile often unpredictably. The authors' approach is to use satellite-based spatial data to retrieve the roughness length $z_0$ and and displacement length d, instead of commonly used tables or even manual assigned parameters, e.g. de- rived from in-situ measurements or 'hand-digitising'. The authors compare their novel satellite-based flow modelling results to predictions based on global land cover maps, lidar scans and manual digitisa- tion. Their goal is to increase the accuracy of wind cross-prediction and reduce uncertainties that come with the traditional methods in cross-prediction in heterogeneous terrain. Their results show a slightly better cross-prediction. The results depend on the complexity of the terrain though. The significance of the paper, however, is not only based on the potential increase of accuracy in cross-prediction, but that the satellite-based approach is using an available and suitable source for (seasonable) land coverage. The authors present a lot of data bases, methods and models aside with an exemplary data evalua- tion. This can be a lot at times and needs to be carefully balanced to keep it to necessary information for the reader. In general the paper is well structured and their methods are well presented. However, there are some (minor) issues with the current state of the manuscript. All issues are listed in the 'Specific comments' section.

*We have reduced the amount of acronyms and databases by removing the sites Alaiz and Perdigao, which were not used in the analysis anyway, except for figure 7. We also grouped the hand-digitized and lidar scan maps in Fig. 7-8 and Table 4, which makes it easier to compare the results. We specified this class also with a different color in the figures, to make it stand out as the 'reference' which we compare the sentinel data to. Tables 1 and 2 have been simplified. All figures have improved labelling to make them stand-alone and easier to understand.*

**Specific comments**

- l.91: Please introduce h properly as the canopy height.

*The canopy height has been introduced.*

- l.130: Here you discuss the plot shown in Fig. 1, especially around the shown LAI around 1. This is where the models differentiate from each other. However, the legend of the plot is overlapping with a large portion

of the plot that is being discussed. This can be solved easily by moving the legend in a way that it does not interfere with the graphs. Alternatively increase the y-axis limits and lift the legend up to around 1.0-1.3 on the y-axis.

*The legend has been moved outside the figure and colours have been adjusted based on comments of reviewer 2.*

- l.134: You mention the differences for different *alpha* and *beta* values in the respective models. You should state in a small statement which one you will use in the study and if you stick to the temperate pine forest. Also add the information where these kind of forest parameters shown in line 134 apply to.

*We added "and we use them throughout this work due to the absence of information on the canopy profile from satellite data."*

- l.256: 'Ryningsnäs is used as an example throughout this section to illustrate our approach'. This is a sentence one would expect at section 3.6. where you introduce the example 'Ryningsnäs'. You could even move Sec. 3.6 into Sec. 4.

*We moved this sentence as suggested.*

- l.296: You introduce the spider-grid analysis or zooming-grid analysis. Can you elaborate why this is used instead of an orthogonal grid as in the later predictions?

*We added an explanation: "The advantage of using a zooming grid is that it concentrates the resolution where it is most needed and we can use arbitrarily distributed points. The latter is for example beneficial for calculating the wind climate at exact positions of wind turbines."*

- l.365: Please introduce *gamma* from Eq. 13.

*Gamma is now introduced.*

- l.398: [. . . ] 'mostly' lead to lower RMS . . . Maybe use more like 'in half of the cases' (6/10). And even then only by a small margin. It would be more representative of the figure.

*We added 'However, at M{'e}rida and Østerild the RMS of $\varepsilon_P$ from the Sentinel based maps is not lower than those from the standard land cover databases.'*

- l.404: Are those $\varepsilon_P$ values averages for all the sites combined, you mention in line 404? Since you mention different sites but only one RMS and method.

*We added the word "combined" to make this more clear.*

- l.405: Can you elaborate why those improvements can not be shown? Or how to understand this con- clusion. Do you mean the data does not show

this? Because for the Cuauhtémoc site the hand- digitised results show a higher RMS. Maybe clear this paragraph up.

*We changed this to: "Thus, averaged over these four sites, satellite-derived estimates of $z_0$ do not yield better power predictions than those based on manually digitized maps."*

- l.411f: Please concretise 'westerly sector', e.g. using the easting that can be used in Fig. 6., as it additional seems that the values for $d$ in average double in the easterly part of the plot (more yellow and green).

*We have changed this to "For the westerly sector (i.e. winds coming from 265-285), $d$ is nearly doubled after reducing the zooming grid to sectorwise displacements (see Fig. 1), which has large implications for the predicted power density (see Fig. 6)."*

- l.413: This is unfortunate as you try to make a case for the Sentinel satellite data, especially a tool for large area with no mast data to validate.

*Yes we agree, but there is still added value of the Sentinel based maps as shown in the rest of the paper, despite these limitations.*

- l.420: Why does Fig. 9 exist? It shows four bars which height difference is not really quantifiable from the graph. The information it is supposed to deliver could be added to Table 4 instead by adding an extra line for the original data base. Or add a column that deals with it.

*We have removed the figure and added the results to Table 4 as suggested.*

- l.453: This should be moved to the conclusions part of the study.

*We think that discussions about future work are more suitable for the discussion.*

**Technical corrections**

- l.165: You refer to Table 3 long before you mention Table 2 in line 206. I would prefer it, if you swap the labelling to avoid confusion.

*We swapped the order.*

- l.420: Wrong cross-reference. I think you mean Fig. 9 not Fig. 3.

*Corrected.*

- l.451: Please spell 'meters' out in this context.

*This has now been spelled out.*

**Reviewer 2**

The authors investigate the ability of satellites to provide roughness length and displacement heights for wind resource assessment. They consider 3 satellite platforms which provide tree height, land cover, and leaf area index. Then, they use 3 forest modules to convert these data to roughness and displacement. They consider these satellite-derived estimates, along with other traditional ways (global land-cover maps; aerial lidar scans; manual digitalization) of obtaining these quantities, and perform power predictions at various sites using WASP. They take each sensor on a met tower and use it to predict the wind speed and power at all other sensors on that tower (vertical predictions) and other towers at the same site (horizontal predictions). A total of 10 sites are considered. The main result is that WASP-derived power predictions obtained with satellite vs lidar land characteristics were comparable (similar error of ~10-11%), indicating that satellite data can be a more affordable alternative to costly aerial lidar campaigns over forested terrain.

**Major comments**

The writing quality is somewhat poor. Subsection 5.3 and Conclusion are the most well- written ones in the manuscript. The rest is certainly readable, but it takes extra effort from the reader to sort through ambiguities and incoherencies. I would expect to be able to focus on the science and results being presented while performing a review, but the writing made it difficult and I spent a substantial amount of time just trying to understand the content that was being presented. Each paragraph should have a clear message and a reason to exist. Figures should stand alone with well-labeled axes and captions, and always show the units. The reader shouldn't have to read the manuscript in detail to understand what the figure is trying to show. Especially since there are so many acronyms for the datasets/models and foreign names for the site locations, it can become difficult to keep track of what's what if the text is not easy to follow. The results presented implicitly include a validation of the WASP models themselves. I suggest that the analysis be reframed to focus exclusively on the effects of what the authors are actually testing: several ways of obtaining $z_0$ and $d$. Either that, or the narrative of the paper should be expanded to include the WASP validation that is being carried out (even if WASP is being employed in a less traditional way, using only one sensor, without the ability to fit to the entire mast profile). It feels like 90% of my time was spent reading the introduction, data, and methodology (19 pages) and then not many results were presented (less than 3 pages). Maybe that's because some results are shown in the methodology? Either way, please consider including some of the results that are not shown (as per manuscript text) and more examples that show spatial variations. It would help the interpretation of the results if the readers could see for themselves the spatial distribution of $z_0$ and/or $d$ at all sites, which could be accomplished

by adding a single figure with subpanels. It might also be helpful to spatially see the progression of data from satellite-derived quantities (h, land cover, lai) to $z_0$ and $d$ for the best-performing and worst-performing sites. Figure 7 is also lacking discussion of some key points: e.g. I don't understand why the hand-digitized map is so bad for one of the MX sites; I don't understand why the proposed method performs worse than MODIS/GLOB/CORINE for Østerild and the two MX sites, and it's not discussed. It seems like Figure 7 can be modified and milked for a much more valuable discussion (and shown more than once, highlighting different aspects of the results, like e.g. vertical vs horizontal predictions). A key, unanswered question is: in what types of sites should one go through the trouble of deriving $z_0$ and $d$ from satellites vs using the global datasets? I would like to see the authors using WASP as they say it's usually used (page 25, lines 483-487) for the sites with two masts: Alaiz, Finland, Perdigao, Sweden, Østerild to get the best prediction possible and then evaluate the effect of different $z_0$ and $d$ estimates. By using WASP the way it's usually used, won't you get results that are more interesting to your audience, and be able to tease out the real sensitivity to these lower-boundary forcing parameters?

*We first of all have largely reorganised the manuscript, moving parts of the methodology to the results section. All figures have been remade with more clearly labelled axes. To produce a lighter version of the methodology we removed the discussions around RIX and the Perdigao and Alaiz sites. As we stated in the first version of the manuscript, one should not use the linearized flow model at these sites. In our original submission we filtered out these sites for most of the plots, but now we removed them altogether from the manuscript. As suggested by the reviewer we added a figure with the roughness length around all the sites. We removed the figure with the tree heights and the databases, because they could be added to table 4 instead. Large parts of results that were presented in the methodology have been moved to the results (mostly former sections 2.3 and 3.6), which resulted in a more balanced paper regarding the methods and results sections.*

*The suggestion to do profile fitting in WAsP was intentionally avoided. We want to focus on the value of an automated model chain as input for the WAsP model.By fitting a profile 'by eye' we introduce a degree of subjectiveness in the paper. Furthermore, the wind shear is largely determined by stability as well, so a good fit of the profile can be caused by a wrong roughness length and stability description that are cancelling each other out. We added this in line 492: "However, such a fitting process can be deceiving, because a good fit is influenced by both $z_0$ and atmospheric stability conditions. It may thus me subject to compensating errors." Here we avoided that by focusing solely on maps that were put into the software without human intervention.*

*We added more information why some sites are not having lower errors. See lines 397-402 in the new manuscript ("The former is a very complex site, that is characterized by forest in all directions. One possible explanation for the higher errors at this site can be a clearing in the forest that is shown to the*

*southeast in the Sentinel maps, but which does not appear in reality. At Østerild the dominating wind directions is from the west, which is characterized by many wind breaks. These are not generally detected in the Sentinel based roughness maps, but contribute significantly to a higher $z_0$. Therefore the background roughness for grassland that was used in the Sentinel maps (see Table A5), might be too low at Østerild."). In any model comparison one cannot expect all sites to perform better, which is why we look at aggregated model performance metrics in this paper. However, we have added hypotheses about why the results where worse at the Østerild site; the poor results of the hand-digitized map at Mexican site was caused by a mislabelled roughness map in our validation script, which has now been corrected.*

*We added a paragraph in the discussion where we highlight at which sites one can expect the largest impact of using tree height maps (lines 444-447 in the new manuscript). "From Fig. 6 it is clear that using the Sentinel maps of tree height rather than standard land cover maps has the highest benefit at sites where masts or turbines are in the middle of the forest (Ryningsnas and Sweden). Taking the displacement height into account for such sites leads to significantly lower $\varepsilon_P$. Because there are seven masts at the Swedish site, leading to a large number of cross-predictions, this site has a large impact on the aggregated results (Table 4).*

**Minor comments**

A very large amount of minor comments is annotated directly onto the pdf

*We appreciate that the reviewer has taken the time to provide concrete suggestions for how the text can be improved. We have taken these suggestions into consideration and adopted most of them in the manuscript. We have also replied directly in the annotated PDF.*

**Reviewer 3**

The authors here present a way to represent wind resource maps using new satellite products from the Sentinel-1 and -2 and the ICESat-2 satellites. They attempt to demonstrate how these new products can be comparable, or even better than, existing best practices such as aerial lidar scan and hand-digitized maps. A large emphasis is placed on the land cover datasets and the forest roughness models used in conjunction with micro-scale flow modeling with WAsP. Ultimately, a cross-prediction is performed against a variety of sites with different roughness/forest characteristics is shown, with the new satellite products outperforming the coarser land cover datasets. Finally, it is emphasized that these new satellite products are not only more accurate for wind resource assessment, but quicker to perform and are continually updated.

In general, the authors do an okay job of presenting their work. Overall, though, it does seem that the delivery of the results is a bit muffled. For example, a great deal of the paper is spent laying out the methods of the work and introducing the different datasets/models. After this, very few pages are concerned with the results. The opportunity does present itself to show more results (especially regarding sites where WaSP performs poorly).

*We have largely restructured the manuscript by moving results that were presented in the methods to the results section. The methodology section has been shortened and made more concise. We discuss in more detail the sites where the model performs poorly (see comments to reviewer 2). We moved the abbreviations column to the left to make it easier to look up the landcover database abbreviations in Table 2. The sites were WAsP is performing poorly (Østerild and Ryningsnäs) are now discussed in detail.We also included roughness maps of all sites to let the reader better understand the discussion in the text.*

**Specific Comments**

- There are a lot of acronyms. I realized this can be hard to avoid, but I found myself constantly jumping around to recall which one meant what. This is done somewhat in Table 3. A simple table in the Appendix would prove beneficial here.

*We have reduced the number of names, by removing some of the sites (Perdigao and Alaiz) and moving the lidar and hand-digitized maps into one class. We moved the abbreviations column to the left to make it easier to look up the landcover database abbreviations in Table 2.*

- Time frames are generally not explained in the figures, such as Figure 6. The satellite time coverage is laid out in Table 3, but it is lost in the text and not explained in the figure captions.

*We now added a reference to the time labels in the figure caption.*

- Line 158: A space is needed in between "in" and "mountainous".

*This sentence has been deleted due to other reviewers' comments.*

- Line 182: What are the consequences of assuming that each subclass of z0 is the same as the class it inherits?

*We added some discussion around this topic: "It is mostly classes with forests that have been split up and one could get a better estimation of $z_0$ by a more detailed analysis of the canopy structure in these subclasses. However, this approach is not attempted here."*

- Line 190-200: I feel this discussion of these machine learning models (Random Forest and Support Vector Regression) came out of nowhere and

hardly any attention is given to the specificities of these. Please elaborate the discussion around these and why they were used in the first place.

*This section has been extended with references and more specific information has been added.*

- Line 387 (Results in general): Results are not shown for the WAsP model in complex terrain with steep slopes, since WAsP is known not to perform well in these conditions. Shouldn't at the very least some of these results be shown? I feel we are missing some of the picture if not.

*We did not show these results because the IBZ model is not applicable there and in fact the results were not used when aggrerating all sites. Instead we therefore removed these sites from the manuscript and instead focus on the sites where WAsP is within its operational envelope.*

- I realize atmospheric stability is not considered in this study, but I do think that would be an interesting addition, especially since we are dealing with various forested sites, and the interactions with these sites and different stabilities could tease out further insights not previously considered. I will leave it up to the authors to include this or not.

*We agree that this is an important topic, but we feel this is more suitable to treat in another paper. In fact, a paper about a new stability treatment in WAsP is currently in preparation where we go into much more detail on this.*

Overall, I think there is a solid paper in here... somewhere. In essence, a greater balance between the methods and results section is needed. Reading 15-20 pages of methods for only ~5 pages of results/discussion is tough. There are some opportunities to include some more results (as explained above), and I believe this could help round out the paper, along with a restructuring of the methods.

*The methods have been largely restructured and the results section has been extended discussing more about the different sites.*